# Human ESCRT-III polymers assemble on positively curved membranes and induce helical membrane tube formation

Aurélie Bertin [1,2,5 ✉], Nicola de Franceschi [1,2,3,5 ✉], Eugenio de la Mora [1,2], Sourav Maiti[4], Maryam Alqabandi[1,2], Nolwen Miguet[3], Aurélie di Cicco[1,2], Wouter H. Roos[4], Stéphanie Mangenot [1,2], Winfried Weissenhorn [3 ✉] & Patricia Bassereau [1,2 ✉]

Endosomal sorting complexes for transport-III (ESCRT-III) assemble in vivo onto membranes with negative Gaussian curvature. How membrane shape influences ESCRT-III polymerization and how ESCRT-III shapes membranes is yet unclear. Human core ESCRT-III proteins, CHMP4B, CHMP2A, CHMP2B and CHMP3 are used to address this issue in vitro by combining membrane nanotube pulling experiments, cryo-electron tomography and AFM. We show that CHMP4B filaments preferentially bind to flat membranes or to tubes with positive mean curvature. Both CHMP2B and CHMP2A/CHMP3 assemble on positively curved membrane tubes. Combinations of CHMP4B/CHMP2B and CHMP4B/CHMP2A/CHMP3 are recruited to the neck of pulled membrane tubes and reshape vesicles into helical "corkscrew-like" membrane tubes. Sub-tomogram averaging reveals that the ESCRT-III filaments assemble parallel and locally perpendicular to the tube axis, highlighting the mechanical stresses imposed by ESCRT-III. Our results underline the versatile membrane remodeling activity of ESCRT-III that may be a general feature required for cellular membrane remodeling processes.

[1] Laboratoire Physico Chimie Curie, Institut Curie, PSL Research University, CNRS UMR168, 75005 Paris, France. [2] Sorbonne Université, 75005 Paris, France. [3] Univ. Grenoble Alpes, CEA, CNRS, Institut de Biologie Structurale (IBS), 71, avenue des Martyrs, 38000 Grenoble, France. [4] Moleculaire Biofysica, Zernike Instituut, Rijksuniversiteit Groningen, Nijenborgh 4, 9747 AG Groningen, The Netherlands. [5] These authors contributed equally: Aurélie Bertin, Nicola de Franceschi. ✉email: aurelie.bertin@curie.fr; N.deFranceschi@tudelft.nl; winfried.weissenhorn@ibs.fr; patricia.bassereau@curie.fr

The Endosomal Sorting Complex Required for Transport-III (ESCRT-III) is part of a conserved membrane remodeling machine. ESCRT-III employs polymer formation to catalyze inside-out membrane fission processes in a large variety of cellular processes, including budding of endosomal vesicles and enveloped viruses, cytokinesis, nuclear envelope reformation, plasma membrane repair, exosome formation, neuron pruning, dendritic spine maintenance, and preperoxisomal vesicle biogenesis[1–8].

Yeast ESCRT-III comprises four subunits Vps20, Snf7, Vps2, and Vps24, which polymerize in this order on endosomal membranes[9], and is dynamically regulated by the ATPase VPSps4[10]. The corresponding human homologues comprise several isoforms named CHMP6 (Vps20), CHMP4A, B, C (Snf7), CHMP2A, B (Vps2), and CHMP3 (Vps24) in addition to CHMP1A, B, CHMP5, CHMP7, and CHMP8/IST1[5]. ESCRT-III proteins adopt an auto-inhibited conformation in the cytosol[11–13], which requires the release of the C-terminal auto-inhibition[14,15]. This leads to the polymerization of loose CHMP4 spirals[16–18], helical CHMP2A-CHMP3 spirals[12,19,20], CHMP2A filaments[21] and Vps24 (CHMP3) filaments[22] in vitro. In vivo, CHMP4 or CHMP2B over-expression leads to membrane tube formation with CHMP4 and CHMP2B filaments inside the tube[23–25]. Polymerization is guided by conformational changes that stabilize the filaments via domain exchange, thereby generating basic surfaces for interaction with positively curved[26] or negatively curved membranes[27] carrying a negative net charge[26,28,29]. Although Snf7 (CHMP4) polymerizes on supported lipid bilayers[18], preformed membrane curvature was suggested to favor Snf7 (CHMP4) membrane interaction[30,31].

Common to all ESCRT-mediated processes is the strict requirement of VPS4 that not only recycles ESCRT-III[32], but actively remodels the polymers in vivo[10,33] and in vitro[34,35]. Furthermore, all ESCRT-catalyzed processes recruit CHMP4 and CHMP2 isoform(s)[2], indicating CHMP4 and CHMP2 are core components for ESCRT-III function. Accordingly, HIV-1 budding requires only one CHMP4 and CHMP2 isoform for virus release[36], although the presence of CHMP3 enhances budding efficiency[21]. This thus suggests a minimal budding/membrane fission machinery that requires CHMP4 and CHMP2 isoforms. Consistent with this proposal, in vitro reconstitution experiments implicated Snf7 (CHMP4), Vps24 (CHMP3) and Vps2 (CHMP2A) in Vps4-driven membrane tube release[29], although Did2 (CHMP1) and IST1 may as well participate in membrane scission[37]. Based on the core fission machinery a number of different models have been proposed to explain ESCRT-catalyzed membrane fission[5,7,38].

In vivo filament assembly has been imaged within bud necks of viruses[36,39,40]. Similarly, ESCRT-III containing spirals have been observed within the cytokinetic midbody[41–43], and proposed to be multi-stranded[33,41]. This thus suggests that ESCRT-III assembles on membranes that exhibit a saddle-like shape with negative Gaussian curvatures (See Supplementary Fig. 1 for a definition of the different curvatures in this work).

Although the intrinsic curvature of the filaments and their flexibility are likely important to shape membranes, the role of ESCRT-III polymers and their preference for different or the same membrane geometries is still unclear[18,20,30,44].

Here we investigate how ESCRT-III polymerization shapes membranes and how it influences their assembly on membranes. To address these questions, we develop in vitro assays based on the essential core of purified human ESCRT-III proteins (CHMP4B, CHMP2A, CHMP2B, and CHMP3) and model membrane systems. We use C-terminally truncated versions of CHMP4B, CHMP2A, and CHMP2B to facilitate polymerization as well as full-length CHMP3. We design confocal microscopy experiments with membrane nanotubes of controlled geometries pulled from Giant Unilamellar Vesicles (GUVs) to study the effect of membrane mean curvature and topology on ESCRT-III protein recruitment and polymerization at the macroscopic scale. Furthermore, by using high-speed AFM (HS-AFM) and cryo-electron microscopy (cryoEM), we obtain nanometer resolution images showing the preferential membrane shape induced upon ESCRT-III assembly on small liposomes and preformed tubes and the corresponding organization of the protein filaments at their surface.

## Results

**CHMP4B does not deform membranes nor sense curvature.** First, we have confirmed with high-speed AFM that the human CHMP4B assembles into spirals when in contact with a negatively charged supported lipid bilayer (SLB) made of 60% DOPC, 30% DOPS, and 10% PI(4,5)P2 (Fig. 1a and Supplementary Movie 1). We measured an average peak to peak distance between filaments within a spiral to $11.3 \pm 1.9$ nm ($N = 134$) (Supplementary Fig. 2), smaller than reported for Snf7 ($17 \pm 3$ nm)[18]. Thus, CHMP4B spirals are built up from one single, unbranched filament, forming a tighter structure than Snf7 spirals that display inter-filament branching[18]. Our ΔC truncated construct might lead to a different molecular organization and thus branching, explaining the difference with Snf7. To test if CHMP4 can deform membranes as a "loaded spring"[45], we have employed an in vitro assay involving deformable vesicles. We have analyzed both the membrane deformation and the organization of CHMP4B filaments by cryoEM. LUVs (Large Unilamellar Vesicles: 50 nm to 1 μm) made of 70% EPC, 10% DOPE, 10% DOPS, 10% PI(4,5)P2 were incubated with 1 μm CHMP4B, plunge frozen and imaged by cryo-EM ($N = 8$ experiments). As displayed in Fig. 1b, CHMP4B assembles into spirals on LUVs. The inter-filament distances within the spirals is $7.8 \pm 2.6$ nm ($N = 208$), corresponding to peak-to-peak distances of $11.3 \pm 2.6$ nm, similar to those measured with HS-AFM. The diameter of the spirals is $193 \pm 63$ nm ($N = 23$). We did not observe any obvious membrane budding or buckling but an apparent flattening as compared to naked LUVs (Supplementary Fig. 3A, B and Supplementary Movies 2A, B). To visualize any 3D deformation, we have performed cryo-electron tomography (cryoET) (Fig. 1c and Supplementary Movie 3) ($N = 5$). When bound to membranes, CHMP4B spirals (in red) are flat without inducing any deformation. (see side view Fig. 1c, bottom and Supplementary Movie 3). They induce a squashing with a height that decreases from about 150 nm for the bare liposomes (Supplementary Movies 2A, B) down to about 50 nm with bound CHMP4B (Supplementary Movie 3). This suggests that the elastic energy stored in the CHMP4B spirals favors a non-curved membrane and no invagination or ex-vagination.

LUVs have a positive Gaussian curvature, in contrast with the negative Gaussian curvature of membranes in biological contexts where ESCRT-III usually localizes (Supplementary Fig. 1a). To study the effect of membrane geometry on ESCRT assembly, we have used assays involving membrane nanotubes with a variety of membrane geometries (Fig. 1d).

We have developed an approach based on laser-triggered fusion[46] that allows ESCRT-III protein encapsulation into negatively charged GUVs[28]. We have encapsulated fluorescently labeled CHMP4B inside a non-charged GUV (cyan) at a concentration of 1 μM and fused it with a GUV containing PI(4,5)P2 (magenta) (Fig. 1e) (for details on lipid compositions, see Methods), leading to a reduction of the charge on the final GUV by 2. A tube was pulled beforehand from the PI(4,5)P2-containing GUV (Fig. 1d, iii), generating a physiologically-relevant membrane geometry for CHMP4B. Tube diameters can be tuned by

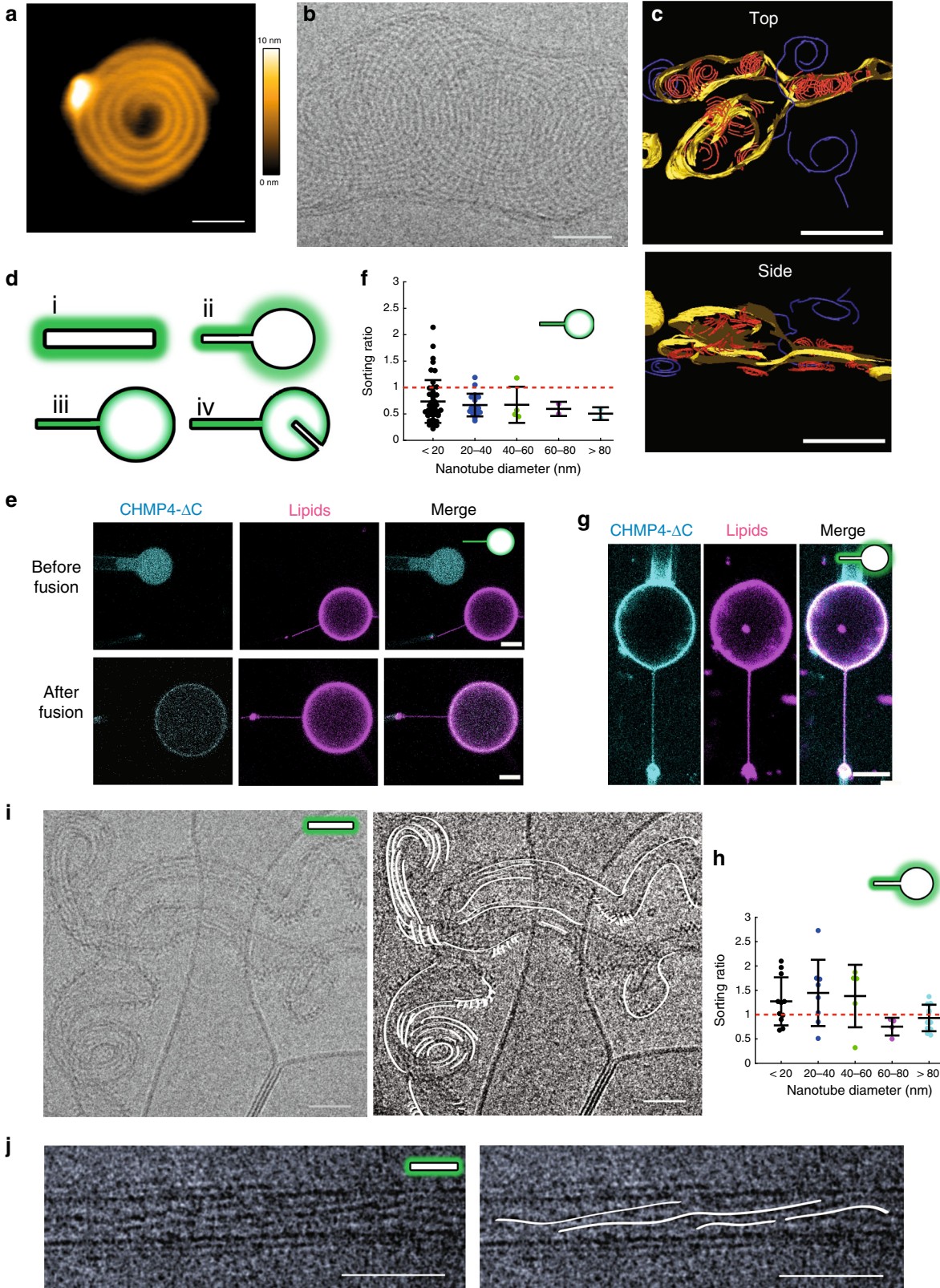

changing membrane tension with a micropipette aspirating the magenta membrane GUV. Using this setup, we observed that CHMP4B was neither enriched at the tube neck, nor in the tube. We calculated the sorting ratio, i.e. the protein enrichment in the tube as compared to the GUV, based on fluorescence[47] ($N = 64$ in total for 17 nanotubes). This ratio is larger than 1 for proteins enriched and lower than 1 for proteins depleted from membrane tubes. This quantification reveals sorting ratios lower than 1 over a large range of tube diameters. Thus, CHMP4B is excluded from tubes with a negative mean curvature (and a null Gaussian curvature) and prefers to bind to the flat surface of the GUV (Fig. 1f). Interestingly, no preferential binding was detected even at diameters corresponding to the expected preferred curvature for Snf7 (between 40 nm and 60 nm)[17,18,20].

**Fig. 1 CHMP4-ΔC flattens LUVs and binds preferentially to flat membranes or to membranes with a positive mean curvature. a** CHMP4B-ΔC spirals observed by HS-AFM on a lipid bilayer. Scale bar: 50 nm. **b** Cryo-EM image of CHMP4B-ΔC spiral on deformable LUVs. Scale bar: 50 nm. **c** Top view (top) and side view (bottom) of a cryo-EM tomogram (Supplementary Movie 2) showing CHMP4B-ΔC spirals (red: CHMP4 filaments polymerized on lipids; blue: filaments polymerized in bulk; yellow: lipids). Scale bar: 200 nm. **d** The different geometries used to study ESCRT-III proteins/membrane interactions. The protein location is indicated by a green shadow. (i) Proteins outside a tube, corresponding to $C > 0$ and $K = 0$. (ii) Proteins outside a nanotube pulled from a GUV: on the tube, $C > 0$ and $K = 0$; on the GUV, $C = K = 0$; and on the neck, $K < 0$. (iii) Proteins inside a nanotube pulled from a GUV: on the tube, $C < 0$ and $K = 0$; on the GUV, $C = K = 0$; and on the neck, $K < 0$. (iv) Spontaneously formed tubule inside a GUV in geometry (iii): on the internal tube, $C > 0$ and $K = 0$. **e** Confocal images corresponding to a GUV fusion experiment in which CHMP4B-ΔC is binding in geometry (iii). Scale bar: 10 μm. **f** Sorting ratio for 17 nanotubes from 17 GUVs in 8 independent GUV preparations and variable diameters (**e**). For each condition, $N$ measurements were made: <20 nm: $N = 19$; 20–40 nm: $N = 28$; 40–60 nm: $N = 11$; 60–80 nm: $N = 6$; >80 nm: $N = 8$. Center line: mean, box limits: SD. Red dashed line: sorting ratio equal to 1. **g** Confocal images corresponding to a GUV fusion experiment where CHMP4B-ΔC binds in geometry (ii). Scale bar: 10 μm. **h** Sorting ratio for 24 nanotubes from 24 GUVs in 10 independent GUV preparations and of variable diameters (**g**). For each nanotube diameter, $N$ measurements were performed: <20 nm: $N = 11$; 20–40 nm: $N = 8$; 40–60 nm: $N = 5$; 60–80 nm: $N = 4$; >80 nm: $N = 11$. Center line: mean, box limits: SD. Red dashed line: sorting ratio equal to 1. **i** Cryo-EM image of CHMP4B polymerized outside deformable membrane nanotubes. Left: Raw image, Right: eye guide. Scale bar: 50 nm. **j** Cryo-EM image of CHMP4B-ΔC filaments polymerized onto non-deformable GlaCer tubes. Left: Raw image, Right: eye guide. Scale bar: 50 nm. **f**, **h** Source data are provided as a Source Data file.

We next tested binding of CHMP4B to positively curved membranes (geometry (ii) (Fig. 1d)). CHMP4B proteins were incubated with GUVs containing PI(4,5)P2 before a membrane nanotube was pulled outwards (Fig. 1g). CHMP4B exhibited a sorting ratio of the order of 1 over the full range of tube diameters ($N = 39$ in total for 24 nanotubes) (Fig. 1h), indicating that CHMP4B can bind to tubes with a positive curvature, although no affinity for this geometry has yet been reported. Moreover, the absence of fluorescence recovery in FRAP experiments on nanotubes after 6 min suggests that CHMP4B forms stable polymers, bound to the tube (Supplementary Fig. 4).

Finally, we used cryo-EM to study the organization of CHMP4B on tubes. Our LUV preparation was generated after a resuspension of a dried lipid film, which preserves PI(4,5)P2 lipids within the bilayer[48]. This methodology generates a heterogeneous suspension of vesicles in size and geometry. 15% ± 3.4% ($N = 315$ vesicles) of the vesicles were spontaneously forming tubular structures in the preparation (arrows in Supplementary Fig. 3A, B). The addition of 1 μM CHMP4B does not induce any further tubulation of the liposomes (20 ± 14.2%, $N = 214$ vesicles). We next analyzed how CHMP4B filaments organize on these preformed tubes. Figure 1i shows that CHMP4B filaments bind to lipid tubes, and align along the main axis of the tubes where the curvature is minimal, forming parallel structures and inducing some helicity to the tubes. We have collected images of the tubes and performed 2D averaging and classification (Supplementary Fig. 3C). 192 sections of tubes decorated by CHMP4 were hand-picked. Five classes were generated by 2D processing. From the averages, repetitive patterns can be discerned parallel to the axis of the tube (Supplementary Fig. 3C, bottom row). We also generated rigid galactocerebroside (GlaCer) nanotubes supplemented with 10% (wt) EPC and 10% (wt) PI(4,5)P2 displaying an external diameter of 25 nm[49]. Similarly, CHMP4B filaments polymerize on these tubular structures and tend to be aligned along the main tube axis (average angle equal to 8.2° ± 5.1° (±SD, $N = 21$), although some twist is visible along the filaments (Fig. 1j).

Altogether, these results do not support previous models of a stiff CHMP4B spiral acting as a loaded spring that could induce membrane bending. Rather, they suggest that CHMP4B, in the absence of the other ESCRT-III proteins, flattens membranes or assembles along the main axis of tubes where the mean curvature is null.

**CHMP2B and CHMP2A/CHMP3 prefer positively curved membranes.** We next analyzed the assembly of CHMP2B and CHMP2A/CHMP3 on membranes with specific geometries. First, we show by HS-AFM that CHMP2B assembles onto SLBs into ring-like structures with a diameter of 16.4 ± 3.1 nm (peak-to-peak distance, $N = 69$) (Fig. 2a). Similar structures have been reported for CHMP2A, assembled in the absence of membranes[19]. In contrast, the in vivo over-expression of CHMP2B induces the formation of rigid tubular membrane protrusions stabilized by CHMP2B helical filaments[25], suggesting that CHMP2B adopts alternative geometries upon binding to membranes. Furthermore, CHMP2B assembles into clusters that localize at the neck of nanotubes pulled from GUVs, but not inside tubes[28]. This suggests that CHMP2B has affinity for negative Gaussian curvature, but not for negative mean curvature, in contrast with in vivo over-expression conditions[25]. In order to test whether CHMP2B binds to positively curved membranes, we incorporated CHMP2B into GUVs and employed the I-BAR protein IRSp53 to form membrane tube invaginations on another set of GUVs[50,51] (geometry (iv) (Fig. 1d)). Fusion of both GUVs (Fig. 2b, left panel) demonstrated that CHMP2B co-localizes with the positively curved membranes of internal tubes (Fig. 2b, right panel) ($N = 7$). This is consistent with an enhanced spontaneous tubulation (30.2 ± 1.6%) observed by cryo-EM after incubation of LUVs with 1 μM CHMP2B. Globally, this thus shows that CHMP2B can assemble on flat membranes and on membranes with a positive mean curvature or a negative Gaussian curvature, which, however, requires the presence of CHMP4 in vivo[25].

CHMP2A and CHMP3 have to be present together for binding negatively charged membranes[19,52]. We thus first studied whether CHMP2A/CHMP3 assembles inside nanotubes, using the geometry (iii) (Fig. 1d). In these experiments, only CHMP2A was fluorescently labeled and CHMP3 was kept unlabeled (see ref. [52]). Proteins were reconstituted inside GUVs at low micromolar concentrations. No binding to the inner leaflet of the membrane nanotube was observed independently of the tube diameter (Fig. 2c, tube diameters equal to 65 and 23 nm). We observe only very weak binding to the flat membrane of the GUV (Fig. 2c). This demonstrates that, CHMP2A/CHMP3 does not assemble on negatively curved membranes and has only a very weak affinity for membranes with a null-curvature under these conditions.

However, in the presence of spontaneously formed internal tubules in the GUVs with a positive mean curvature (geometry (iv) Fig. 2d), we noticed a strong enrichment of the proteins on these structures. Conversely, we observed with cryo-EM that these proteins generate positive membrane curvature since the fraction of tubular structures is increased as compared to the control (31 ± 5%) when CHMP2A (0.5 μM) and CHMP3 (3 μM) are added.

In order to further confirm the preference of CHMP2A/CHMP3 for positively curved membranes, we incubated GUVs with CHMP2A/CHMP3 and pulled a nanotube outwards

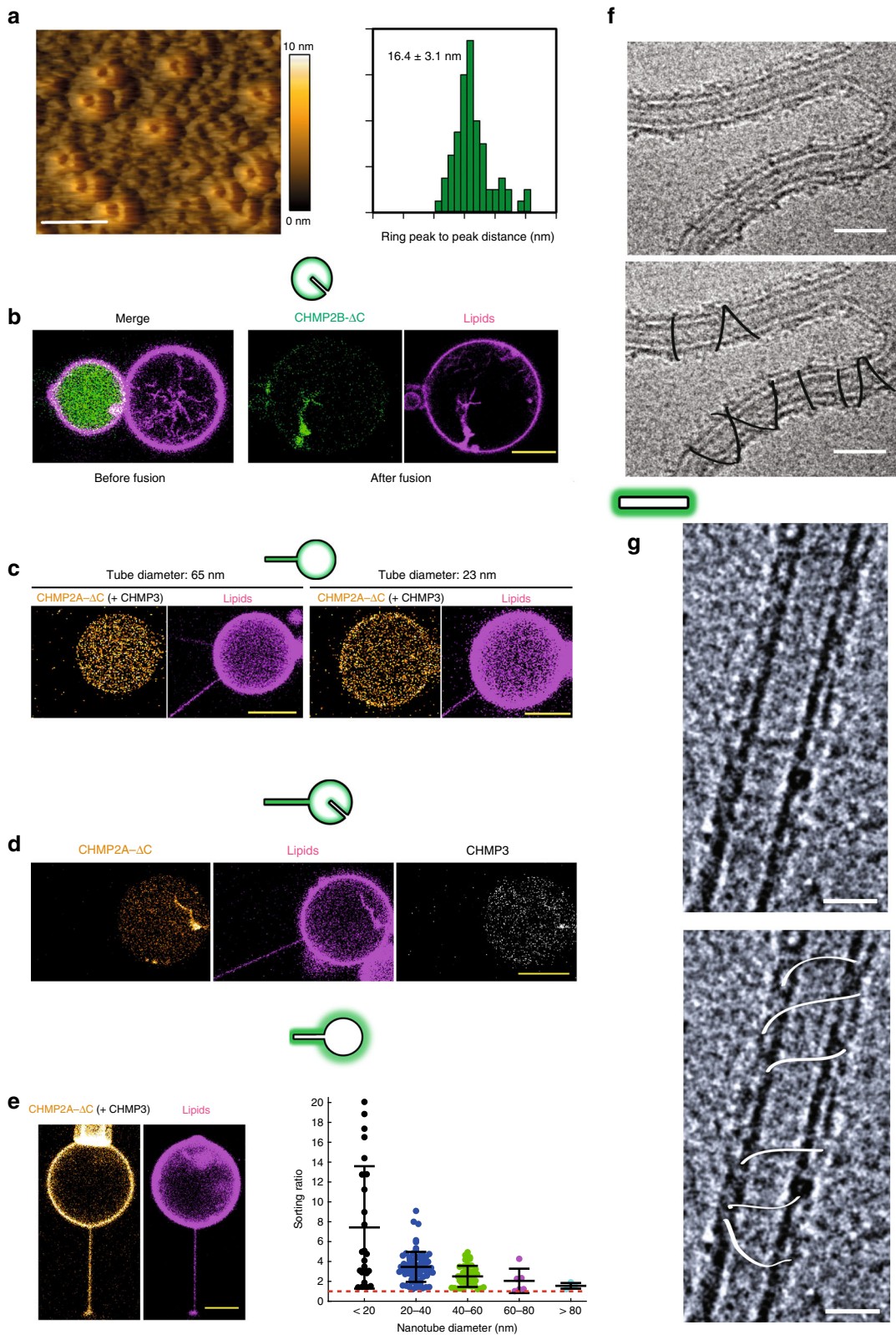

(geometry (ii)). Here, CHMP2A associates in the presence of CHMP3 with the GUV membrane as well as the outer leaflet of the pulled membrane tube (Fig. 2e). By quantifying the sorting ratio in this geometry, we confirmed that CHMP2A/CHMP3 bind to membrane with a positive mean curvature ($N = 145$ in total, 24 nanotubes) (Fig. 2e, right). Moreover, since the sorting ratio increases with tube curvature (the inverse of the radius) up to about 5 for tube diameters smaller than 20 nm, it demonstrates that the CHMP2A/CHMP3 complex can polymerize on membrane in a positive curvature-dependent matter, similarly to CHMP2B.

In agreement with these tube pulling experiments, Cryo-EM visualization of the CHMP2A/CHMP3 assembly onto tubular membrane structures (geometry (i); Fig. 1d) revealed a helical

**Fig. 2 CHMP2B-ΔC and CHMP2A-ΔC/CHMP3 essentially assemble on positively curved tubes. a** HS-AFM image of CHMP2B-ΔC rings on a flat, non-deformable SLB. The quantification of ring diameters is shown. **b** Confocal images corresponding to a GUV fusion experiment in which CHMP2B-ΔC is exposed to a geometry (iv) induced by the I-BAR domain of IRSp53 (non-fluorescent), tubulating the membrane when present on the exterior of the GUV. $N = 7$. Scale bar: 10 μm. **c** Confocal images corresponding to a GUV fusion experiment in which CHMP2A-ΔC + CHMP3 are binding in geometry (iii). CHMP3 is unlabeled. Left GUV: tube diameter = 65 nm. Right GUV: tube diameter = 23 nm. Scale bars: 10 μm. **d** Confocal images corresponding to a GUV fusion experiment in which CHMP2A-ΔC + CHMP3 are binding in geometry (iv), showing the affinity of the assembly for internal positively curved tubes. Scale bar: 10 μm. **e** Left: Confocal images corresponding to a GUV fusion experiment in which CHMP2A-ΔC + CHMP3 are binding in geometry (ii). CHMP3 is unlabeled. Scale bar: 10 μm. Right: Quantification of the sorting ratio for 24 nanotubes of variable diameters from 25 GUVs in 9 independent GUV preparations. For each nanotube diameter, $N$ measurements have been performed: <20 nm: $N = 20$; 20–40 nm: $N = 74$; 40–60 nm: $N = 41$; 60–80 nm: $N = 6$; >80 nm: $N = 4$. Center line: mean, box limits: SD. The red dashed line corresponds to a sorting ratio equal to 1. Scale bar: 10 μm. **f** Cryo-EM image of CHMP2A-ΔC/CHMP3 filaments polymerized outside deformable membrane nanotubes. Top: Raw image, Below: Guide for the eyes. Scale bar: 50 nm. **g** Cryo-EM image of CHMP2A-ΔC/CHMP3 filaments polymerized outside non-deformable GlaCer tubes. Top: Raw image, Bottom: Guide for the eyes. Scale bar: 50 nm. **a**, **e** Source data are provided as a Source Data file.

loose polymer wrapping around membrane tubes perpendicularly to the main tube axis, both on spontaneous deformable tubes (Fig. 2f) and on rigid GlaCer tubes (Fig. 2g). This further demonstrates the affinity of CHMP2A/CHMP3 for membranes with a positive mean curvature, which is different from the linear arrangement of CHMP4B along the main axis of tubes (Fig.1i, j).

*CHMP4B/2B and CHMP4B/2A/3 bind inside nanotubes' neck.* In vivo, ESCRT-III complexes function on membranes with a negative Gaussian curvature. We therefore co-encapsulated CHMP4B and CHMP2B (both fluorescent) as well as CHMP4B and CHMP2A/CHMP3 (with fluorescent CHMP4B and CHMP2A) at low micromolar concentrations in EPC GUVs and fused them with GUVs containing PI(4,5)P2 from which a tube was pulled (Fig. 1d, geometry iii).

Upon fusion, in some cases, no membrane binding was detected, probably due to a too low protein concentration. When binding occurs, both CHMP4B and CHMP2B bind to the inner leaflet of the GUV with a local enrichment of CHMP4B and CHMP2B at the nanotube neck (Fig. 3a) in 66% of the cases ($N = 12$, 4 experiments). No protein was detected inside the nanotubes. When internal tubular structures were present inside the GUV (geometry iv) (4 GUVs), both proteins were found to be bound to these tubes.

In the case of CHMP4B/CHMP2A/CHMP3, slightly different results were observed. When membrane binding was observed, only CHMP4B was detected on the GUVs ($N = 15$, 9 experiments). In about 30% of the cases, however, we could detect a local enrichment of both CHMP4B and CHMP2A at the neck of the nanotube (Fig. 3b). All the proteins were always excluded from the interior of the nanotubes. Eventually, CHMP4B was also often strongly bound to internal tubular structures (Supplementary Fig. 5) (90% of 10 GUVs), with only weak enrichment, if any, of CHMP2A on these tubules.

Altogether, we have found that in the absence of other ESCRT partners, these minimal complexes can be recruited to the neck of membrane tube structures exhibiting a negative Gaussian curvature. In addition, they have some affinity for membranes with a positive mean curvature.

*CHMP4B/2A/3 and CHMP4B/2B reshape vesicles into pipe surfaces.* The complex Vps2/Vps24 was shown to induce deformation of regular Snf7 spirals assembled on non-deformable flat SLBs[33]. We show here with HS-AFM that CHMP2B has a very similar effect on CHMP4B spirals on a SLB (Fig. 3c). The CHMP4B spirals lose their regularity upon addition of CHMP2B (Supplementary Fig. 6 and Supplementary Movie 4). Similar observations were also obtained with cryo-EM on flattened LUVs (Supplementary Fig. 8A).

We next studied in real time by HS-AFM whether membrane reshaping occurs upon ESCRT-III protein addition. In this assay, small unilamellar vesicles (SUVs) (diameters of 60-100 nm) were

immobilized on a mica surface before CHMP4B was added followed by CHMP2B. The addition of CHMP4B (2 μM) did not change the spherical shape of the SUVs even after 15 min (Supplementary Fig. 7A, B). However, further addition of CHMP2B (1 μM) induced the formation of an outward protrusion from the vesicle (Fig. 3d, Supplementary Fig. 7C and Supplementary Movies 5A, B) over the same time period, showing that CHMP4B with CHMP2B can mechanically deform SUVs.

These deformations were analyzed with cryo-EM over longer time on LUVs. LUVs were incubated for 1 h with CHMP4B (0.5–1 μM) and upon addition of either CHMP2B (0.5–1 μM, $N = 15$ experiments) or CHMP2A (0.5–1 μM)/CHMP3 (1.5–3 μM) ($N = 13$ experiments), for one additional hour induced, extensive vesicle tubulation was observed (Fig. 3e and Supplementary Fig. 8B). Close to 100% of the LUVs were tubulated under these conditions. This concentration range is optimal since no extensive tubulation is observed below, and protein aggregates form above. Strikingly, both CHMP4B/CHMP2B and CHMP4B/CHMP3A/CHMP3 remodel the vesicles into helical tubes, like a corkscrew (a geometrical shape called a pipe surface) (Fig. 3f, h). Helical membrane tube formation by CHMP4B/CHMP2B was confirmed by cryoET (Fig. 3g). Both CHMP4B/CHMP2B and CHMP4B/CHMP2A/CHMP3 helical membrane tubes revealed parallel filaments, following the tube axis (Fig. 3f, h, i). The sequence of protein addition is essential to trigger such membrane reshaping. When CHMP4B and CHMP2B, or CHMP2A/CHMP3, are added simultaneously, or when CHMP4B is added after CHMP2B ($N = 2$ experiments) or CHMP2A/CHMP3 ($N = 2$ experiments), the helical tubular deformations no longer occur and only flat spirals, like for CHMP4B-only (Fig. 1b), are observed (Supplementary Fig. 8 C, 8D and 8E),. Hence, CHMP4B has to assemble first on liposomes to nucleate the helical membrane tube deformation by either CHMP2B or CHMP2A/CHMP3.

Figure 3k schematizes the pipe surface of the membranes upon binding of the ESCRT-III, where $w$ is the width of the spiral, $\phi$ the diameter of the tube and $d$ the distance between adjacent parallel filaments on the tube. We found that the width $w$ is conserved and equal to $115.1 \pm 16.2$ nm ($N = 66$) for CHMP4B/CHMP2B and $110.9 \pm 20$ nm ($N = 66$) for CHMP4B/CHMP2A/CHMP3. The diameters of the tubular structures are displayed in Fig. 3l. The diameters of the tubes decorated by CHMP4B only ($\phi = 37.4 \pm 14$ nm, $N = 43$) are significantly larger than the tubes induced in combination with CHMP2B ($\phi = 26.2 \pm 4.4$ nm, $N = 107$) or CHMP3/CHMP2A ($\phi = 27.9 \pm 11$ nm, $N = 233$), suggesting that the final organization of the proteins on the membrane constricts the tubes. In contrast, the distance between filamentous structures parallel to the tube axis is similar for CHMP4B/CHMP2B samples ($d = 4.2 \pm 1.1$ nm ($N = 60$)), for CHMP4B/CHMP2A/CHMP3 samples ($d = 4.4 \pm 1.2$ nm ($N = 66$)) and for CHMP4B only ($d = 4.4 \pm 0.8$ nm ($N = 31$)) (Fig. 3m). In addition, striations

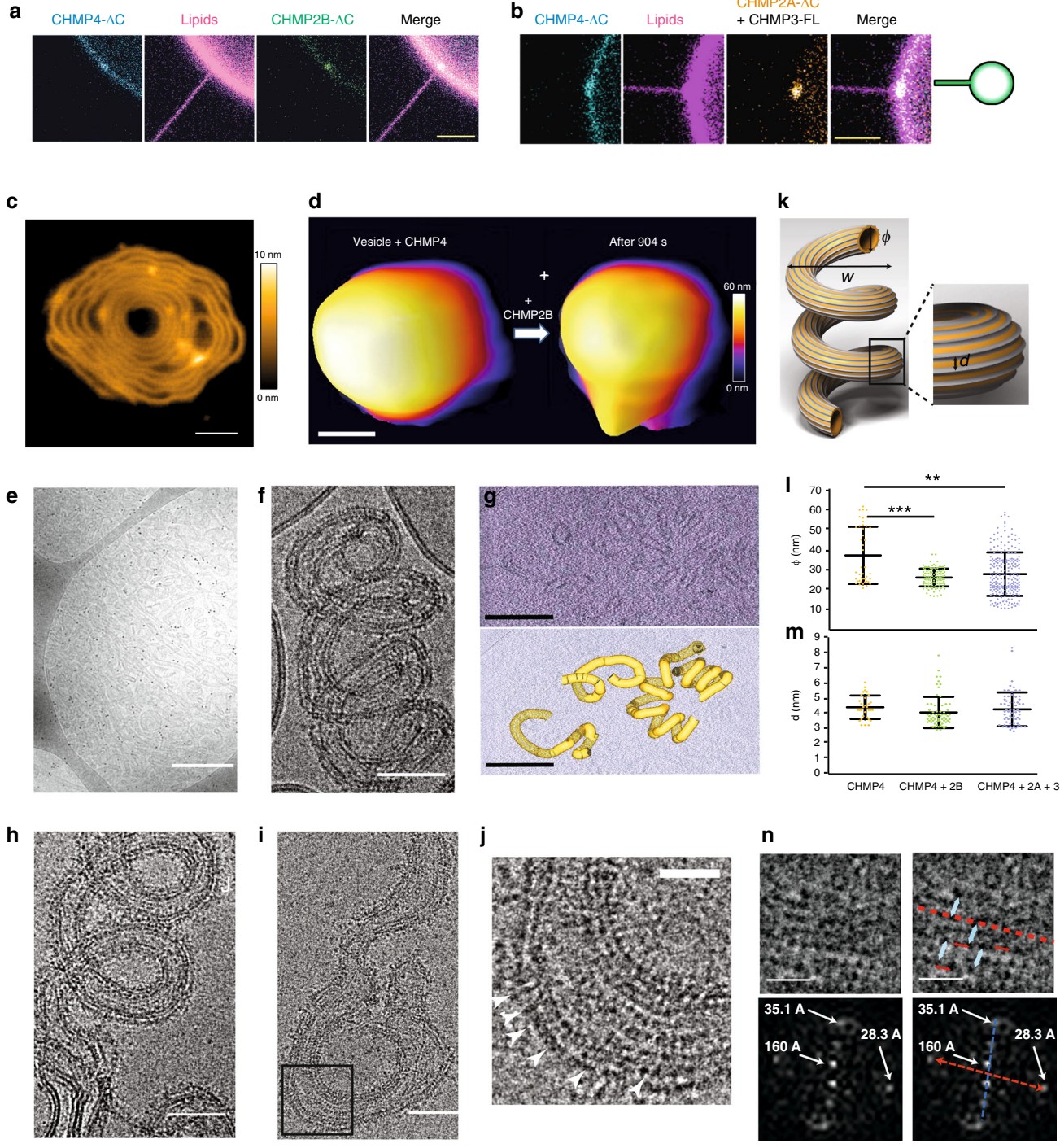

perpendicular to the long axis of the tubes are also present (Fig. 3j, arrows). To further quantify this observation, we have performed 2D classification out of 350 sub-portions of tubes for the CHMP4B/CHMP2B sample (Fig. 3n, top). The 10 resulting classes and the resulting Fourier transformation show unambiguous evidence for periodical structures both along the axis of the tubes (red line) and perpendicular to the tube (blue line, 20% of the dataset) (Supplementary Fig. 8F). In Fourier space, all classes generated peak densities along the direction related to the tube axis (red) (Fig. 3n, bottom and Supplementary Fig. 8F), corresponding to a repetitive pattern of consecutive filaments with a mean distance of 3.5 ± 0.3 nm, averaged from the 10 classes, similar to our image analysis (Fig. 3m). In addition, diffraction peaks indicating a repeated distance perpendicular to

the axis of the tubes (blue) (Fig. 3n) were also present for 20% of the obtained classes (3 classes) (Supplementary Table 1). We obtained 3.2 ± 0.4 nm. Note that no such orthogonal structures were detected with CHMP4B only (Supplementary Fig. 3C). Hence, two sets of perpendicular filaments are bound to the pipe surface: a first set of parallel filaments along the main axis of the helical tube and locally, on about 20% of the total tube length, a second set of perpendicular filaments, possibly influencing its diameter (Fig. 3l).

To uncover the arrangement of the ESCRT-III proteins at nanometer scales, we carried out cryoET, followed by sub-tomogram averaging, of CHMP4/CHMP2B-induced tubular structures. Four different populations of filaments decorating the pipe-like architecture could be identified. Note that each tube

**Fig. 3 Complexes are recruited inside tube necks and reshape liposomes into helical tubes. a** Representative confocal images of GUVs in geometry (iii): preferential recruitment of CHMP4B-ΔC and CHMP2B-ΔC inside the tube neck. Scale bar: 5 μm. **b** Same as (**a**): preferential recruitment of CHMP4B-ΔC and CHMP2A-ΔC(+ CHMP3) inside the tube neck. Scale bar: 5 μm. **c** Effect of CHMP2B-ΔC addition to a CHMP4B spiral on a flat SLB, imaged by HS-AFM. Scale bar: 50 nm. **d** Deformation of a small liposome first incubated with CHMP4B-ΔC, after addition of CHMP2B-ΔC imaged by HS-AFM (snapshots from Movie S5A). Scale bar: 100 nm. **e** Effect of CHMP2B-ΔC addition to liposomes pre-incubated with CHMP4B-ΔC, imaged by Cryo-EM at low magnification. Scale bar: 500 nm. **f** Details of a helical tube structure induced by CHMP4B-ΔC + CHMP2B-ΔC. Scale bar: 50 nm. **g** Cryo-EM tomogram of a helical tube structure formed by CHMP4B-ΔC + CHMP2B-ΔC (Top). Scale bar: 200 nm. Bottom: segmentation of the tubular membrane (yellow) from the cryo tomogram above. Scale bar: 200 nm. **h, i** Helical tubes induced by CHMP4B-ΔC + CHMP2A-ΔC + CHMP3. Scale bar: 50 nm. **j** Zoom corresponding to the black frame in Fig. 3i. Scale bar: 20 nm. **k** Scheme of the helical tubes. $w$: width of the spiral, $\phi$: diameter of the tube and $d$: distance between parallel filaments. **l** Measurement from the Cryo-EM images of the tube diameters $\phi$ for CHMP4B-ΔC ($N = 43$), CHMP4B-ΔC/CHMP2B-ΔC ($N = 107$, $p = 4.5 \times 10^{-6}$) and CHMP4B-ΔC/CHMP2A-ΔC/CHMP3 ($N = 233$, $p = 7.4 \times 10^{-5}$). Center line: mean, box limits: SD. **m** Measurement from the Cryo-EM images of the distance $d$ between filaments parallel to tube axis for CHMP4B-ΔC ($N = 31$), CHMP4B-ΔC/CHMP2B-ΔC ($N = 66$) and CHMP4B-ΔC/CHMP2A-ΔC/CHMP3 ($N = 32$). Center line: mean, box limits: SD. **n** Top: Class-average of helical tube sections (sub-class #3 in Supplementary Table 1) formed by CHMP4B-ΔC/CHMP2B-ΔC. Blue arrows: distance between 2 structures parallel to the tube axis; red arrows: distance between 2 structures perpendicular to the tube axis (See Supplementary Fig. 8F). Scale bar: 10 nm. Bottom: Fourier-Transform (FT) with the distances corresponding to the Bragg peaks. Left: Raw data. Right: The red line represents the direction of the tube axis and the blue line to the perpendicular direction along the tube section. **l, m** Source data are provided as a Source Data file.

exhibits exclusively one type of filament arrangement and no mixed arrangements coexist within the same "pipe". The first, comprising one third (34%) of the analyzed structures, does not display any organized patterned architecture. The second (Fig. 4a) is composed of individual protein filaments that decorate the pipes in a homogeneous fashion. This group was the predominant ordered architecture observed in the sample (33%). Indeed, after averaging lipid tubes using their whole sections (Supplementary Fig. 9A), we observed that 14 filaments decorate the tubular structure in a periodic and regular pattern, independent of the curvature (Supplementary Fig. 9A, Supplementary Movies 6 and 7). The reconstruction was obtained at a resolution of 26.1 Å (See FSC curve in Supplementary Fig. 9B). The third population consists of paired filaments that were found in only 3% of the dataset (Fig. 4b) and its final average was determined at a resolution of 26.1 Å (see FSC, Supplementary Fig. 9B) (Supplementary Movie 8), Interestingly, multireference analysis (MRA), resulted in different classes grouped according to their distribution along the pipe (Supplementary Fig. 9C: class 1 and class 2). Class 1 corresponds to negatively curved portion of tubes (inner side), where filaments are scarce (Supplementary Fig. 9D). Class 2 corresponds to filaments bound to the outer side of tubes (positive curvature) with a higher density, suggesting that paired ESCRT filaments have a higher affinity for positively curved membranes. A similar asymmetric distribution has been described in a recent report on yeast ESCRTs[53]. The fourth population, comprising the remaining 30% of the dataset, is composed of filaments bridged by protein connections perpendicular to the main axis of the tubes (see Fig. 4c, arrows) (Supplementary Movie 9). Its final average was determined at 28.3 Å resolution. This set of bridging proteins is most likely related to the striations perpendicular to the main axis of the lipid tubes visualized in Fig. 3j, highlighted by diffraction spots perpendicular to the main axis of lipid tubes (Fig. 3n).

Taken together, our analyses demonstrate that the observed macroscopic tubulation into a corkscrew-like architecture is driven by distinct nanometer ultra-structures of ESCRT filaments.

## Discussion

Membrane remodeling by ESCRT-III polymers implicates in many cases negative Gaussian membrane curvature. It has been shown that ESCRT-III assembles at or inside bud necks of endosomal vesicles[10], enveloped viruses[36,39,40] and within the cytokinetic midbody[41,43]. Less is known about membrane shape requirements or effects of ESCRT-III recruitment during the other ESCRT-catalyzed membrane remodeling processes. Interaction with positively curved membranes has been so far only

reported for ESCRT-III CHMP1B that forms helical structures on membrane tubes in vitro and in vivo on endosomal tubular extensions[54–56]. Our results reveal that neither CHMP4B, CHM2A/CHMP3 nor CHMP2B have on their own affinity for membranes with a mean negative curvature. Instead, we show that CHMP4B assembles into spirals on flat membranes similar to yeast Snf7[18]. However, CHMP4B alone does not deform membranes. Polymerization into spiral structures on the positively curved membranes of LUVs only leads to membrane flattening, but no membrane buckling occurs under these conditions in contrast to theoretical predictions[18,45], suggesting that the filaments made of CHMP4B have a low bending rigidity (see also[53]). Upon incorporation of CHMP4B inside GUVs with membrane tubes pulled outwards, CHMP4B does not concentrate at the tube as observed at the bud neck under CHMP2 double knockdown HIV-1 budding conditions[36]. Thus, functional CHMP4B recruitment to membrane necks may require prior assembly of ESCRT-I and ESCRT-II complexes or Alix that coordinate the assembly of CHMP4B filaments[57,58] or enhance CHMP4B affinity for the membrane, rather than CHMP4B having a preference for certain membrane curvatures[30,31]. In contrast, CHMP4B assembles on the outside of membrane tubes, thereby forming filaments parallel to the tube axis where mean membrane curvature is zero. The large-scale twisting of the CHMP4B-decorated flexible membrane tubes further supports the helical nature of the CHMP4B filaments.

Strikingly, we have shown that both CHMP2A/CHMP3 and CHMP2B do not assemble inside tubes but rather polymerize on the outer side of membrane tubes. Moreover, high positive curvature enhances CHMP2A/CHMP3 polymerization that forms helical filaments wrapping around tubes, perpendicular to the tube axis. Although the structure of the polymers appears more lose than for CHMP1B[26], it establishes that not only CHMP1B interacts with positive curved membranes, but also ESCRT-III core members. Notably, the latter have been implicated in vivo in membrane remodeling with an opposite membrane geometry.

Only the combination of CHMP4B/CHMP2B or CHMP4B/CHMP2A/CHMP3 is found occasionally enriched at the neck of membrane nanotubes consistent with the proposal that a CHMP4 polymer forms a platform for downstream ESCRT-III assembly[9,36]. In addition, CHMP2A-CHMP3 and CHMP2B systematically remodel CHMP4B-bound LUVs into regular helical tubes with pipe surfaces. Three-dimensional helical polymeric structures have also been reported in the absence of membranes for Snf7, Vps2 and Vps24[20,59], and for CHMP2A/CHMP3[19]. For both CHMP4B/CHMP2B or CHMP4B/CHMP2A/CHMP3, the width $w$ of the ESCRT-III helices is in the

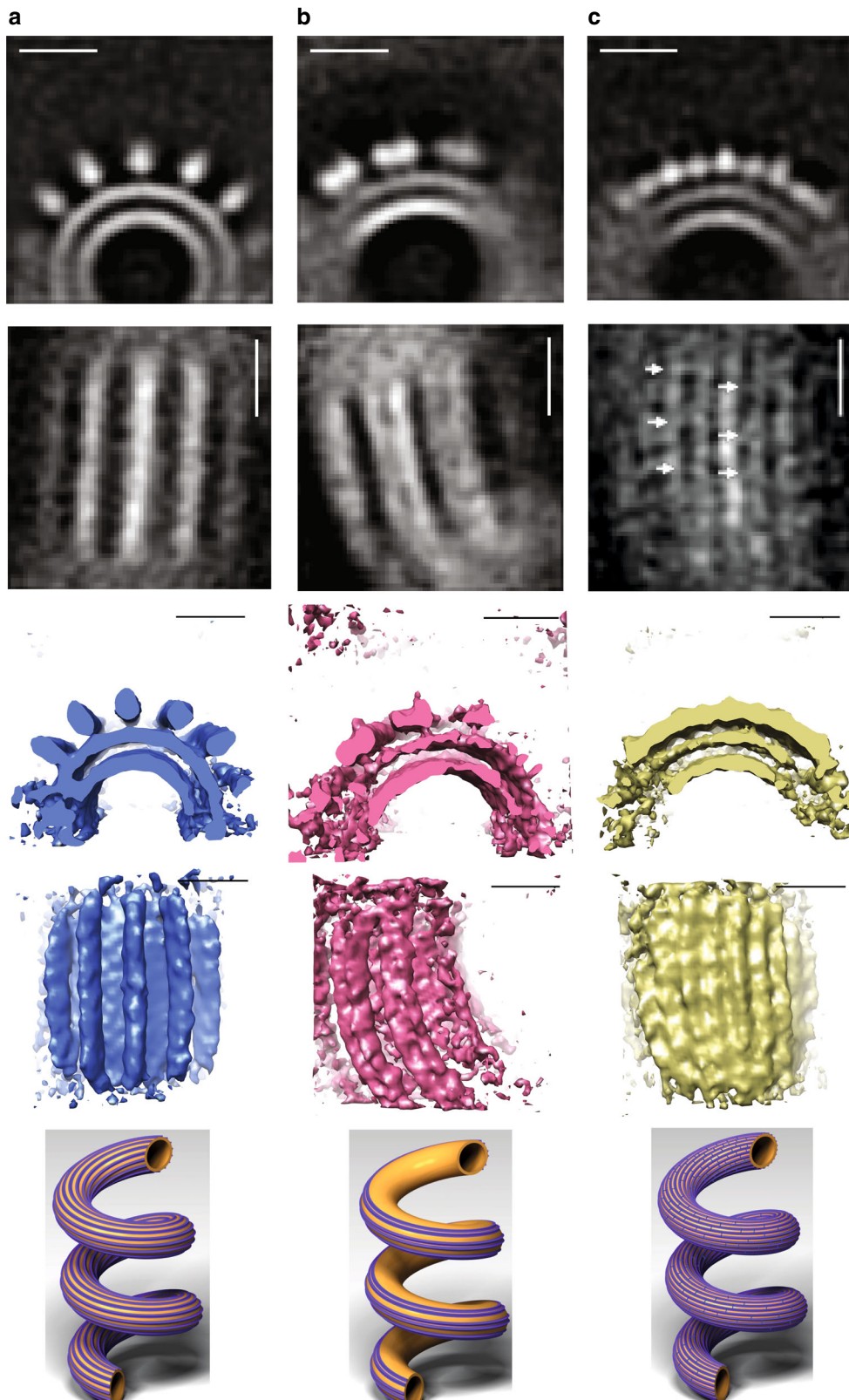

**Fig. 4 Sub-tomogram averaging of CHMP4B-ΔC/CHMP2B-ΔC decorated pipes.** Populations of ESCRT filaments bound to tubular membranes resulting from sub-tomogram averaging. Upper line: orthoslices viewed from the cross sections of tubes. Second line: orthoslices viewed from the top of tubes. Third line: 3D reconstructions viewed from the cross sections of tubes. Fourth line: reconstructions viewed from the top of tubes. Bottom line: schematic representations of the CHMP4B-ΔC/CHMP2B-ΔC-decorated pipes. Scale bars: 10 nm. **a** Single ESCRT individual filaments bound to lipid tubes. Inner tube diameter: 14.8 nm, outer tube diameter: 21.7 nm. Protein density diameter: 30.2 nm. Reconstruction from 1721 particles. **b** Paired ESCRT filaments bound to lipid tubes. Inner tube diameter: 15.4 nm, outer tube diameter: 21.7 nm. Reconstruction from 524 particles. **c** High density of filaments bound to lipid tubes. Inner tube diameter: 18.5 nm, outer tube diameter: 25.4 nm. Reconstruction from 381 particles. Arrows point to structures perpendicular to the tube axis.

order of 110 nm, thus about twice larger than the preferred one for Snf7[18,20] or CHMP4B[23,24]. These tubular structures have a smaller diameter than the occasional tubes observed with CHMP4B alone, suggesting particular mechanical properties of the mixed polymers. The mechanism behind such a massive membrane remodeling is yet unclear. Using sub-tomogram averaging, we could sort the architecture of the CHMP assemblies on the pipe surfaces at nanometer scales in two main categories, single filaments and filaments with perpendicular connections, and an additional minority made of double filaments, although all leading to the same macroscopic membrane geometry. This indicates that only a limited set of filaments may be required to induce this helical tubular structure. The resolution of the current structures does not permit to conclude whether the filaments are formed by open or closed CHMP conformations[11,12,26,27]. Using budding yeast ESCRT-III proteins, a paper in the same issue of the journal reports that Snf7, VPS2 and Vps24 (10 μM: 5 μM: 5–10 μM) forms similar corkscrew-like membrane tube structures[53], suggesting that this remodeling capacity of ESCRT-III is conserved among species. However, in this report, the global membrane shape transformation was essentially attributed to filament doublets non-homogeneously distributed around the tube, with different adhesion energy depending on the face in contact with the membrane. The authors propose that the possibility for the filaments to tilt and roll on the membrane for optimizing their binding can generate torque on the filament axis that can produce constriction and scission[60]. This type of structure, however, represents only a very minor fraction of the organizations that we have observed with human ESCRTs, suggesting that other mechanisms can also shape vesicles in to pipe surfaces. In contrast, on at least one third of the surface of the spiraled tubes, we observe a combination of filaments aligned along the helical tube axis crosslinked by orthogonal structures. This sort of scaffold that combines both trends of CHMP4B to form a wide spiral organization and of CHMP2A/CHMP3 to wrap around tubes, can explain the emergence of the pipe surface geometry for the membrane, although what sets the tube diameter is unclear. The respective protein compositions of the perpendicular structures might also be different, but at this stage we cannot distinguish them. In addition, a large fraction of the surface is covered with single filaments regularly distributed around the tube diameter, without any apparent connection between them. It is possible that the connections between filaments are too scarce or too disorganized to be detected. Nevertheless, altogether our results and those from von Filseck et al[53] show that the pipe shape constitutes a robust membrane deformation that occurs over a large range of protein concentrations, with proteins from different species.

Although the helical tubular shape of the membrane may be far from membrane geometries where these core ESCRT-III proteins have been localized in vivo, it nevertheless reveals the mechanical stresses that these protein assemblies exert on membranes. The pipe surface represents the membrane shape that minimizes the mechanical energy of the system in the absence of any external constraints. It also shows the capacity of the proteins to assemble onto an "outside-in" geometry, similarly to CHMP1B and IST1[5]. Nevertheless, when overexpressed in human cells, CHMP4B induces tubules with an "in-outside" geometry[5,23]. In vivo, CHMP4B is recruited to the membrane via CHMP6, which in turn is recruited by ESCRT-II and ESCRT-I or by Alix[1,2,5,7]. The ESCRT-III proteins may be then forced to assemble in a non-optimal geometry and in return exert mechanical forces on the neck structure to release frustration. How these reciprocal interactions combined to the protein turn-over due to the VPS4 ATPase and the possible constrictive action of CHMP1 and IST1[37,56] lead to membrane scission remains to be established.

Here we provide novel insight on how mechanics and geometry of the membrane and of ESCRT-III assemblies can generate forces to shape a membrane neck.

## Methods

**Reagents**. Common reagents were purchased from VWR reagents. L-α-phosphatidylcholine (EPC, 840051P), 1,2-dioleoyl-sn-glycero-3-phosphocholine (DOPC, 850375P), Cholesterol (700000P), 1,2-dioleoyl-sn-glycero-3-phosphoethanolamine (DOPE, 850725P), 1,2-dioleoyl-sn-glycero-3-phospho-L-serine (DOPS, 840035P), L-α-phosphatidylinositol-4,5-bisphosphate (PI(4,5)P2, 840046P), 1,2-dioleoyl-sn-glycero-3-phosphoethanolamine-N-(biotinyl) (PE-Biotin, 870282P) and 1,2-dioleoyl-sn-glycero-3-phosphoethanolamine-N-(lissamine rhodamine B sulfonyl) (Rhod-PE, 810150P) were purchased from Avanti polar.

Gold nanorods Streptavidin-conjugated gold nanorods (C12-10-850-TS-DIH-50) were purchased from Nanopartz™.

Streptavidin-coated polystyrene beads (diameter 3.2 μm) for the tube pulling experiments were purchased from Spherotech.

**Recombinant proteins**. CHMP3 (full length) was expressed in Escherichia coli BL21 cells (New England BioLabs, # C2530H) for 3 h at 37 °C[11]. Briefly, cells were harvested by centrifugation (4000g for 20 min at 4 °C) and the bacterial pellet was resuspended in 50 ml of binding buffer A (20 mM Bicine pH 9.3, 300 mM NaCl, 5 mM imidazole, 1% CHAPS/1 mM PMSF). The bacteria were lysed by sonication and CHMP3-FL was purified by Ni²⁺ chromatography. A final gel filtration chromatography step was performed in buffer B (20 mM Hepes pH 7.6, 150 mM NaCl).

CHMP2A-ΔC containing residues 9–161 was expressed as MBP-fusion protein in Escherichia coli BL21 cells[61] for 1 h at 37 °C. Cells were harvested by centrifugation (4000g for 20 min at 4 °C) and the bacterial pellet was resuspended in 50 ml of binding buffer C (20 mM Hepes pH 7.6, 300 mM NaCl, 300 mM KCl). The bacteria were lysed by sonication, and CHMP2A-ΔC was purified on an amylose column. CHMP2A-ΔC was labeled overnight at 4 °C with Alexa Fluor 405 NHS Ester (Thermo Fisher Scientific) using a molar ratio (Alexa Fluor:protein) of 2:1. A final gel filtration chromatography step was performed in a buffer B. CHMP3-FL and CHMP2A-ΔC were concentrated to 20 μM, and immediately frozen in liquid nitrogen with 0.1% of methyl cellulose (Sigma-Aldrich) as cryoprotectant. All aliquots were kept at −80 °C prior to experiments.

CHMP2B-ΔC, containing amino acids 1–154 and a C-terminal SGSC linker for cystein-specific labeling[62], was expressed in Escherichia coli BL21 cells for 4 h at 37 °C. Cells were lysed by sonication in buffer D (50 mM Tris-HCl pH 7.4, 1 M NaCl, 10 mM DTT and protease inhibitor (Complete EDTA free, Roche) at the concentration indicated by the manufacturer) and the soluble fraction was discarded after centrifugation (50,000g, 20 min, 4 °C). The pellet was washed three times with buffer E (50 mM Tris-HCl pH 7.4, 2 M urea, 2% Triton X-100 and 2 mM β-mercaptoethanol). The last wash was performed in absence of urea and Triton X-100. The extraction of CHMP2B was performed in 50 mM Tris-HCl pH 7.4, 8 M guanidine, 2 mM β-mercaptoethanol overnight at 4 °C. After centrifugation (50,000g, 20 min, 4 °C), CHMP2B was purified by Ni²⁺-chromatography in buffer F (50 mM Tris-HCl pH 7.4, 8 M urea). The protein was eluted in 50 mM Tris-HCl pH 7.4, 8 M urea, 2 mM β-mercaptoethanol, 250 mM imidazole. Refolding was performed by rapid dilution of CHMP2B into buffer G (50 mM Tris-HCl pH 7.4, 200 mM NaCl, 2 mM DTT, 50 mM L-glutamate, 50 mM L-arginine) and a final concentration of 2 μM. CHMP2B was concentrated by passing it over a Ni²⁺ column in buffer H (50 mM Tris-HCl pH 7.4, 200 mM NaCl) and eluted in buffer I (50 mM Tris-HCl pH 7.4, 300 mM NaCl, 250 mM imidazole). CHMP2B was labeled overnight at 4 °C with Alexa Fluor 488 C5 Maleimide (Thermo Scientific) with a molar ratio (Alexa Fluor:protein) of 2:1. A final gel filtration chromatography step was performed on a superdex75 column in buffer J (50 mM Tris-HCl pH 7.4, 100 mM NaCl). The protein was concentrated to 20 μM, and immediately frozen in liquid nitrogen with 0.1% of methyl cellulose (Sigma-Aldrich) as a cryo-protectant. All aliquots were kept at −80 °C prior to experiments.

CHMP4B-ΔC, containing residues 1–175 followed by a linker of 20 aa (SNSASDDASASASASADEDASS) and CHMP4B residues 204–224, was expressed as MBP fusion protein[16] in Escherichia coli BL21 cells for 2 h at 37 °C. Cells were harvested by centrifugation (4000g for 20 min at 4 °C) and the bacterial pellet was resuspended in 50 ml of binding buffer K (50 mM Hepes pH 7.6, 300 mM NaCl, 300 mM KCl). The bacteria were lysed by sonication. The CHMP4B protein was purified on an amylose column. CHMP4B was labeled overnight at 4 °C with Alexa 555 succimidyl ester or 633 succimidyl ester (Thermo Fisher Scientific) using a molar ratio (Alexa Fluor:protein) of 2:1. A final gel filtration chromatography step was performed in the buffer KJ. CHMP4B were concentrated to 15 μM and immediately frozen in liquid nitrogen with 0.1% of methyl cellulose (Sigma Aldrich) as cryo-protectant. All aliquots were kept at −80 °C prior to experiments.

**Cryo-electron microscopy sample preparation and imaging**. A lipid mixture (70% EPC, 10% DOPE, 10% DOPS, 10% PI(4,5)P2) at 1 mg mL⁻¹ was quickly dried under argon for 2 min and next under vacuum for 30 min. LUVs (Large Unilamellar Vesicles) of variable size (50–500 nm) were obtained by resuspension

and vortexing of the lipid film after addition of a buffered solution to reach a final concentration of 0.1 mg mL$^{-1}$. Different combinations of CHMP proteins were incubated with the vesicles at room temperature for 1 h. We used CHMP4B, CHMP2B and CHMP2A at a concentration of 0.5 or 1 μM, and CHMP3 at a concentration of 1.5 or 3 μM, in the buffer 25 mM Tris HCl pH 8, 50 mM NaCl, containing the AcTEV protease (Thermofisher scientific) in excess. For the protein mixtures, we used a ratio of 1:3 for CHMP2A/CHMP3, 1:1 for CHMP4B/ CHMP2B, and 1:1:3 for CHMP4B/CHMP2A/CHMP3, respectively. A 4 μL drop of the solution was deposited on a glow discharged lacey carbon electron microscopy grid (Ted Pella, USA). Most of the solution was blotted away from the grid to leave a thin (less than 100 nm) film of aqueous solution. The blotting was carried out on the opposite side from the liquid drop and plunge frozen in liquid ethane at −181 °C using an automated freeze plunging apparatus (EMGP, Leica, Germany). The samples were kept in liquid nitrogen and imaged using three different microscopes. A Tecnai G2 (FEI, Eindhoven, Netherlands) Lab$_6$ microscope operated at 200 kV and equipped with a 4k × 4k CMOS camera (F416, TVIPS) was used (Institut Curie) at a magnification of 50,000 with a pixel size of 2.13 Å and a dose per image of 15 electrons per Å$^2$. Some of the imaging was performed as well on a 200 kV FEG microscope equipped with a direct detector (Falcon camera) (Institut Pasteur) at a 50,000 magnification with a pixel size of 2 Å and a dose per image of 15 electrons per Å$^2$. A 300 kV FEG (Field Emission Gun) POLARA microscope (FEI, Eindhoven, Netherlands) equipped with an energy filter and a direct detector (K2 camera, Gatan) was also employed (IBS, Grenoble). In this case, the imaging was performed at a magnification of 81,000 with a pixel size of 1.21 Å using a movie mode collecting 40 successive frames for a total dose of 50 electrons per Å$^2$. The different frames were subsequently aligned.

**Cryo-electron tomography.** The samples were prepared as described above. 10 nm size gold beads were added to the solution before being plunge frozen. Tilt series were collected in low dose mode, every two degrees, using a Tecnai G2 (FEI, Eindhoven, Netherlands) microscope operated at 200 kV and equipped with a 4k × 4k CMOS camera (F416, TVIPS) (Institut Curie). To preserve the information and minimize irradiations at low tilt angles, the following angular scheme was applied: from 0° to 34°, then from −2° to −60° and finally from 36° to 60°. The dose per image was 0.8 electrons per Å$^2$. The imaging was performed at a magnification of 50,000 and each image was binned twice for a final pixel size of 4.26 Å. The consecutive images were aligned using the IMOD software suite[63]. Back projection was performed using IMOD and SIRT reconstruction was carried out using Tomo3d. The segmentation was performed manually using IMOD.

The dataset used for sub-tomogram averaging consisted of 28 tilt-series collected at the Electron Microscopy Core Facility of the European Molecular Biology Laboratory (EMBL) in Heidelberg. Image acquisition was performed on a Titan Krios microscope (FEI) operated at 300 kV using a Quantum post-column energy filter and a Gatan K2 Summit direct detector controlled by SerialEM[64]. Tilt-series were collected using the dose-symmetric scheme (Hagen et al. 2017) in the range of ±60° and a 3° angular increment and a defocus range between −1.5 and −4.25 μm. Tilt images consisted of 13 super-resolution frames with a total dose per tilt-series of 140 e$^-$ Å$^{-2}$. Alignment of tilt images based on gold fiducials, CTF estimation (CTFPlotter) and CTF correction (CTFPhaseFlip) were achieved using the IMOD suite[63,65]. Tomograms were reconstructed by the weighted back-projection method applied over the CTF-corrected aligned stacks in IMOD. Bin 4×(pixel size = 5.3 Å) tomograms were reconstructed using the SIRT-like filtering method with 50 iterations to facilitate identification of membrane bilayers during catalogue annotation. Protein-induced tubes were identified and annotated in Dynamo[66,67] as filaments around axis. Tube radius was determined upon aligning and averaging particles cropped from the tube axis as the center of sub-volumes of 88 pixels (46.6 nm) using as alignment mask a cylinder of 22 nm radius. Then, the center of the box was displaced to the tube's membrane surface and the selected oversampling geometry was 16 cropping points per radius separated by 6 pixels along the tube axis. Reference free sub-tomogram averaging was performed on sub-volumes of 34$^3$ nm in Dynamo.

**Lipid mixture preparation for GUVs.** Lipid stock solutions were mixed at a total concentration of 1 mg/ml in chloroform with following molar ratio: 54.7% EPC; 10% DOPS; 10% DOPE; 15% cholesterol; 10% PI(4,5)P2; 0.2% DSPE-PEG2000-Biotin; 0.1% PE–Rhodamine for the charged GUVs and 98.8% EPC, 0.2% DSPE-PEG2000-Biotin for the non-charged GUVs (containing encapsulated proteins).

**Protocol for GUVs preparation.** GUVs were prepared with the PVA gel-assisted swelling method as previously described[68]. Briefly, PVA gel (5% Poly(vinyl alcohol)), 50 mM Sucrose, 25 mM NaCl and 25 mM Tris-HCl, at pH 7.4) was deposited on plasma cleaned (PDC-32G, Harrick) glass coverslips (18×18 mm, VWR International, France) and dried for 50 min at 60 °C. 15 μl of lipid solutions at 1 mg/ml were deposited on the PVA-coated slides and residual solvent was removed under vacuum for 20 min at room temperature. The lipid film was then rehydrated at room temperature for 45 min with the appropriate GUV growth buffer: 100 mM sucrose and 50 mM NaCl for the experiments with proteins in the external medium, and buffers listed in the Supplementary Tables 2–5 for the experiments with encapsulated proteins.

**GUVs with proteins in the external medium.** All proteins have been incubated with GUVs at a concentration of 500 nM in a buffer containing 100 mM glucose, 25 mM Tris pH 7.4 and 50 mM NaCl for 30′ together with 10 nM TEV, which was sufficient to cleave at least 90% of the MBP tags in 15′ at room temperature (not shown).

**GUVs with encapsulated proteins.** All proteins were co-encapsulated with purified recombinant TEV protease at a final concentration of 10 nM, which was sufficient to cleave at least 90% of the MBP tags in 15′ at room temperature (not shown). Growth and observation buffers have been adjusted (Supplementary Tables 2–5) to each specific protein or proteins combination in order to balance the osmotic pressure. The final protein concentration and ratios between proteins results from the balance between a number of factors, such as maintaining a final NaCl concentration of ~50 mM after fusion (each protein is stored in a different storage buffer and at a different concentration). This is similar to the experiments with protein binding outside of the GUVs, maintaining a sufficient amount of sucrose inside the GUV to allow for its sedimentation in the observation chamber and finally avoiding protein inhibition. Indeed, in co-encapsulation experiments, CHMP4B binding to membrane was inhibited by CHMP2A if CHMP4B/CHMP2A ratio was raised above ~2:1, in line with previous work showing capping activity of Vps2 towards Snf7[33].

Practically, CHMP4B stock solution being at 300 mM NaCl + 300 mM KCl, after addition of about 60 μl of CHMP4B, the encapsulation mixture has a salt concentration of ~100 mM. Thus, after fusion with a PI(4,5)P2 vesicle of equal size, the salt concentration drops to ~50 mM.

For, CHMP2A and CHMP3, both stock solutions are at 150 mM NaCl. After addition of 80 μl of CHMP2A and 50 μl of CHMP3, the encapsulation mixture has a NaCl concentration of ~90 mM. After fusion with a PI(4,5)P2 vesicle of equal size, the NaCl concentration drops to about 45 mM.

After addition of 50 μl of CHMP4B, 30 μl of CHMP2A and 20 μl of CHMP3, the encapsulation mixture has a NaCl concentration of about 100 mM. After fusion with a PI(4,5)P2 vesicle of equal size, the salt concentration drops to ~50 mM.

Similarly, CHMP4B and 50 μl of CHMP2B, the encapsulation mixture has a NaCl concentration of ~100 mM. After fusion with a PI(4,5)P2-containing vesicle of equal size, the salt concentration drops to ~50 mM.

The final protein concentrations in the GUVs after fusion are listed in Supplementary Table 6.

The fusion procedure was performed as previously described[28] (see ref. [69] for a detailed protocol). Briefly, for each experiment, two types of GUVs extracted from each PVA slide were mixed with the relative external buffer matching the osmolarity and centrifuged for 10 min at 1000g. GUVs taken from the bottom of the Eppendorf were incubated with gold nanorods 20 min at room temperature and then added to the imaging chamber.

Gold nanorods Streptavidin-conjugated gold nanorods have a peak of absorption at λ = 834 nm, with a tail spanning the wavelength of the infrared laser of the optical tweezers (λ = 1064 nm). The stock solution (typical concentration 1750 ppm) was diluted 1:100 upon incubation with GUVs and again diluted 1:40 when GUVs were transferred to the observation chamber.

Fusion of GUV pairs coated with the gold nanorods is achieved by bringing the GUVs hold by two micropipettes into close contact with micromanipulation and by locally heating the nanorods by focusing the infrared laser on the contact through the objective.

**Tube pulling experiments.** The tube pulling experiments and analysis have been performed as described in detail in refs. [47,69]. Briefly, for experiments probing the affinity of the proteins for positive curvature, the tube was formed by bringing briefly the GUV coated with proteins in contact with a streptavidin-coated bead trapped with the optical tweezer and moved away. For experiments involving encapsulation and fusion, the tube was pulled from the PI(4,5)P2-containing GUV prior to fusion, using a streptavidin-coated bead hold by a third micromanipulator. Fusion was then performed between the GUV pair, keeping the membrane tube in place.

The values of tube diameter (in nm) were deduced from the lipid fluorescence intensities the tube in comparison with the fluorescence in the GUV, after calibration:

$$\text{Tube diameter} = 2 \times 200 \times \frac{I_{\text{tube}}^{\text{lipid}}}{I_{\text{GUV}}^{\text{lipid}}} \qquad (1)$$

where $I_{\text{tube}}^{\text{lipid}}$ and $I_{\text{GUV}}^{\text{lipid}}$ represent the fluorescence intensities of the lipids in the tube and in the GUV, respectively.

The sorting ratio S (protein enrichment in the tube) was calculated using

$$S = \frac{\frac{I_{\text{tube}}^{\text{protein}}}{I_{\text{GUV}}^{\text{protein}}}}{\frac{I_{\text{tube}}^{\text{lipid}}}{I_{\text{GUV}}^{\text{lipid}}}} \qquad (2)$$

where $I_{\text{tube}}^{\text{protein}}$ and $I_{\text{GUV}}^{\text{protein}}$ represent the fluorescence intensities of the proteins in the tube and in the GUV, respectively.

**High-speed AFM**. All HS-AFM data were taken in amplitude modulation mode using a sample scanning HS-AFM [Research Institute of Biomolecule Metrology (RIBM), Japan]. Short cantilevers (USC-F1.2-k0.15, NanoWorld, Switzerland) with spring constant of 0.15 N/m, resonance frequency around 0.6 MHz, and a quality factor of ~2 in buffer were used. The cantilever-free amplitude is 1 nm (3 nm for imaging liposomes), and the set-point amplitude for the cantilever oscillation was set around 0.8 nm (2.7 nm for liposomes). Unless mentioned, all the HS-AFM recordings were performed in buffer containing 25 mM Tris pH 7.4 and 50 mM NaCl.

The HS-AFM experiments on supported lipid bilayers were performed with SLBs composed of 60% DOPC, 30% DOPS, and 10% PI(4,5)P2. The SLBs were formed by incubating LUVs on top of freshly cleaved mica, as described in ref. [35]. Briefly, LUVs were thawed at room temperature and diluted to a concentration of 0.2 mg/ml in buffer (25 mM Tris, pH 7.4, 50 mM NaCl). Then the LUVs were incubated onto the freshly cleaved mica for 5–10 min, and rinsed with the same buffer afterwards. After formation of SLBs, the surface was imaged without addition of protein. While imaging the SLB, the proteins were added to the AFM liquid chamber to reach a final concentration of 2 μM for CHMP4B, and 1 μM for CHMP2B. The formation of CHMP4B spirals on SLBs occurred within 10 minutes after incubation. To capture the effect of CHMP2B on CHMP4B spiral, CHMP2B was only added after the formation of CHMP4B spirals was confirmed by HS-AFM imaging.

The HS-AFM experiments for dynamic membrane deformation were performed using liposomes (SUVs) composed of 50.7% EPC; 10% DOPS; 10% DOPE; 15% cholesterol; 10% PI(4,5)P2; 0.2% DSPE-PEG2000-Biotin; 0.1% PE–Rhodamine. The SUVs were obtained by sonicating a LUV mixture for 30 s. The SUVs were incubated for 5 min on freshly cleaved mica, and imaged under HS-AFM. Then, CHMP4B was added to reach a final concentration of 2 μM in the chamber. Later, CHMP2B (at a final concentration of 1 μM) was added, but only after confirmed spiral formation (typically after 10 min of CHMP4B addition) on randomly formed membrane patches on mica surface. All the HS-AFM images were processed with Igor Pro with a built-in script from RIBM (Japan), and ImageJ software. Unless otherwise mentioned, all reported values are presented as mean ± SD.

**Reporting summary**. Further information on research design is available in the Nature Research Reporting Summary linked to this article.

## Data availability

Data supporting the findings of this manuscript are available from the corresponding authors upon reasonable request. A reporting summary for this Article is available as a Supplementary Information file. The source data underlying Figs. 1f, h, 2a, e, 3l, m, Supplementary Fig. 2C, D, Supplementary Fig. 6C–E are provided as a Source Data file. One example tomogram as well as our sub-tomogram averages have been deposited in the EMBD, with the accession code: EMD-10720.

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

## Acknowledgements

The authors thank Daniel Levy for support and insightful discussions, Michael Henderson for carefully reading the manuscript. We thank Eric Nicolau for his drawing skills. This work was initiated with a grant from FINOVI (W. W., P. B.) and was supported by the ANR (ANR-14-CE09-0003-01) (W.W., P.B.), by the Institut Curie and the Centre National de la Recherche Scientifique (CNRS). W.W. acknowledges the Institute Universitaire de France (IUF) and the platforms of the Grenoble Instruct-ERIC center (ISBG; UMS 3518 CNRS-CEA-UGA-EMBL) within the Grenoble Partnership for Structural Biology (PSB). Platform access was supported by FRISBI (ANR-10-INBS-05-02) and GRAL, a project of the University Grenoble Alpes graduate school (Ecoles Universitaires de Recherche) CBH-EUR-GS (ANR-17-EURE-0003). For cryo-electron microscopy we acknowledge the support of G. Schoehn at the Grenoble FRISBI/Instruct-ERIC electron microscopy platform, F. Weiss and W. Hagen of the cryo-electron microscopy platform of the European Molecular Biology Laboratory (EMBL, Heidelberg) and G. Pehau-Arnaudet, M. Nilges from the UBI facility (Institut Pasteur, Paris). The Falcon II detector at the UBI facility was financed by the "Equipement d'excellence CACSICE" and the Grenoble Instruct-ERIC EM platform acknowledges support from the FRM and GIS IBiSA. Access to the EMBL cryo-electron microscopy facility was supported by iNEXT (project number 653706), funded by the Horizon 2020 program of the European Union. We further acknowledge the Cell and Tissue Imaging (PICT IBiSA, Institut Curie) platform supported by France-BioImaging (ANR10-INBS-04). N.D.F was funded by post-doctoral fellowships from the Institut Curie, the Fondation pour la Recherche Médicale and Marie Curie actions (MSCA-IF-2016 #751715 (ESCRT model)). E.M.L was supported by a post-doctoral fellowship from ANR (ANR-15-CE11-0027-02). M.A. was funded by the Université Pierre et Marie Curie/Sorbonne Université, Doctoral school "Physique en Ile de France" (ED-564) and the Fondation pour la Recherche Médicale. P.B. is a member of the CNRS consortium CellTiss. P.B., S.Man, A.B., A.C. are members of the Labex CelTisPhyBio (ANR-11-LABX0038) and Paris Sciences et Lettres (ANR-10-IDEX-0001-02).

## Author contributions

P.B. and W.W. conceived the study. A.B., A.C., S.Man. performed cryo-EM experiments and A.B., E.M.L. analyzed data. N.F. performed tube pulling experiments and analyzed data. W.R. supervised HS-AFM work and S.Mai performed HS-AFM experiments and analyzed data. M.A. optimized the characteristics of the membrane samples. N.M. purified and labeled CHMP proteins. P.B., W.W., and S.Man equally contributed to this work. A.B., N.F., S.Mai., W.R., S.Man., W.W., and P.B. wrote the paper.

## Competing interests

The authors declare no competing interests.
