## [Peer Review File · Nature Communications]

Reviewers' comments:

Reviewer #1 (Remarks to the Author):

This paper by Bertin et al characterizes how ESCRT-III polymer assembly depends on local curvature of the membrane. The ESCRT field is a highly populated and - as far as I am aware - also quite contentious one, and some of the authors' conclusions appear in direct contradiction with previous claims by others. Following the editor's request, I herewith primarily comment on AFM aspects of the study, but not on the overall significance of the results, as that would be outside my area of expertise.

Overall, the manuscript is clearly written, but could be clarified in the Results section, lines 145-149, which describes to sorting ratios larger than 1 and smaller than 1, with references to Fig. 1F: Fig. 1F only shows sorting ratios smaller than 1. This text would benefit from clarification on what are expectations/predictions and what are results.

Re AFM, the AFM results mostly appear complementary and not strictly essential to demonstrate the authors' points. The most intriguing observation is that of the formation of an outward protrusion at a vesicle upon addition of CHMP2B and CHMP4B, and the fact that they appear to show an intact vesicle on a mica substrate, whereas such vesicles tend to collapse on mica substrates. It would be helpful if the authors could comment (in main text of methods) what prevents the vesicles from collapsing on the mica. In addition, with such rather intriguing AFM results, it would be beneficial if an additional supplementary movie could be included, as a repeat to show the same (or a similar) result as acquired in an independent experiments.

Reviewer #2 (Remarks to the Author):

The paper by Bertin et al studies binding and assembly of ESCRTIII polymers on membranes of different curvatures. The study finds that CHMP4B prefers flat membranes, and in fact can flatten curved membranes, but cannot deform the membrane on its own. CHMP4B/CHMP2B and CHMP4B/CHM2A/CHMP3 both prefer positively curved membranes and can deform the membrane into positively curved helical tubes.

The structure and function of ESCRTIII polymers is currently a highly active field and this paper presents a very nice and timely addition to the field, which will be surely highly cited. The paper is clear and well written, and its results align with the recent findings of several other groups that are currently published as preprints, in particular Nguyen et al 10.1101/798181, Filseck et al 10.1101/716308, and Pfitzner et al 10.1101/718080. The previous papers deal with yeast ESCRTIII and CHMP1B+IST1, hence the current work is highly complementary. I would recommend publishing it in Nat Comm, but I would recommend addressing several points before:

- Can the authors say something about the stoichiometry of the proteins in these polymers? I think that would be extremely useful. Based on the previous work (eg Pfitzner et al), it is likely that the exact composition (including the stoichiometry) of the filament determines its curvature sensitivity and function. Hence could it be that the same combination of proteins, but at a different ratio, forms polymers that have a different curvature preference? Have the authors tested different relative concentrations? I think it might be misleading to claim that a combination of 2 proteins always does something, without specifying their ratio.
- In a similar paper Filseck et al comment how the membrane-binding face of filaments might be different between different polymers. Can the authors say something about this here?
- Can the authors say something about the rigidity of these filaments, to attempt to explain the pipe structure?

- It is interesting that Vps4 is not needed for the creation of deformation and the production of mechanical force. Can the authors comment on what its role could be?
- In my opinion the authors did an excellent job in writing the paper in a clear manner, summarising the findings at the end of the every section. However, I believe it is missing a connection with the currently published work. In particular with the Nguyen et al and Pfitzner et al. The community would greatly benefit from a comment on how it ties together.
- Along the same lines, it would be good to connect the more recent theoretical findings where the tense spiral hypothesis was questioned (Harker-Kirschneck et al, BMC Biology).
- In their conclusions the authors say this work provides new insight into how ESCRTIII assemblies generate force. But to me that is not obvious from this work, can the authors clarify this?

Reviewer #3 (Remarks to the Author):

I am serving as a technical reviewer for the cryo-electron tomography (cryo-ET) and subtomogram averaging experiments presented in this paper. As I am not an expert on ESCRT biology I cannot judge the novelty and the impact of the paper on the field.

The authors use a number of biophysical methods including cryo-ET to describe the interplay between polymerization of ESCRT-III components and membrane shape.

It is good practice in the cryo-EM field to deposit reported data for public access after publication. The authors should deposit at least one example tomogram and all of their subtomogram averages in the Electron Microscopy Data Bank (EMDB) and include the accession codes in the paper.

I appreciate that the authors visualized central aspects of their cryo-ET data in the form of very informative movies.

There are several aspects including the analysis and interpretation of the cryo-ET experiments that should be addressed in a revised version of the manuscript.

1) Major points:

- Using AFM, the authors measure a peak-to-peak distance of filaments in CHMP4B spirals that is 50% smaller than what was observed using the same methods for the yeast homolog Snf7. Since this is not just 'slightly smaller' (line 110), the authors should discuss possible reasons.
- The authors suggest that CHMP4B filament spirals flatten vesicles. This is a very problematic conclusion, because isolated vesicular structures always appear significantly flattened in cryo-ET data, most likely because they are squeezed by the surface tension of the water/buffer film during the blotting process that is part of the cryo-EM grid preparation. There are numerous published cryo-ET studies that document this effect. In the methods section, the authors even state that vesicles are typically embedded in "a thin (less than 100 nm) film of aqueous solution", which would clearly compress them in Z-direction. While I agree that CHMP4B seems to colocalize with flat membrane areas, there is no experimental support that CHMP4B actually induces the flattening. It appears more likely that the flattening is a result of the surface tension and CHMP4B preferentially co-localizes with these areas. The authors should discuss this possibility in the manuscript and possibly tone down their claim that CHMP4B filaments are the cause for flattened vesicles.
- How can the authors judge vesicle flattening in 2D projection images as shown in Fig. S3A,B and where are the control vesicles in Fig. S3A,B that the authors must have compared the CHMP4-decorated vesicles to in order to come to their conclusion?
- In the CHMP2A/CHMP3 binding experiment, the authors specifically state that 'proteins were

reconstituted inside GUVs at nearly physiological concentrations'. What about concentrations in the other experiments?

- And related: CHMP concentration seems relevant to membrane binding (line 259). How much does the concentration bias the experiments?
- The authors claim that the diameter of CHMP4B-decorated vs. CHMP4B-CHMP2B-decorated tubes is significantly larger (line 317). At least judging from the size of the error bars in Fig. 3L, this seems not to be the case.
- I think the entire subtomogram averaging section of the paper could benefit immensely from some further analysis. For instance, single filament subunits are not resolved along the tube axis and the bridges in Fig. 4C are not convincing. Given that the raw data acquired at EMBL in Heidelberg was likely of very high quality, one possible cause for the rather low resolution obtained during subtomogram averaging could be limited long-range order, in particular regarding the distance between individual filaments. Therefore, the authors could try to focus processing on smaller segments using tighter masks e.g. excluding not directly adjacent filaments.
- Related to the previous comment: The FSC in Fig. S8B never goes down to zero! This should never happen and needs to be resolved. The most likely cause for this FSC behavior is the presence of duplicate particles in the dataset, which result in artificially high correlation and thus overestimated resolution.

2) Minor points:

- In many of the electron micrographs or tomograms, the contrast is rather low and the features discussed in the text are hardly visible. I guess the authors are showing 'raw' unfiltered data. The contrast of features could be improved by averaging several Z-slices or by filtering the data, e.g. with a low-pass filter, a Wiener filter or denoising algorithms.
- Does the behavior of CHMP4B in the binding experiments in Figs. 1E,F differ from what has been observed for the yeast homolog Snf7? If so, could the authors repeat some of the binding experiments with Snf7 to confirm the different behavior?
- The authors claim that CHMP4B filaments "tend to be aligned along the main tube axis". Could the authors confirm this in a more objective manner for instance by plotting a histogram of the angular difference between CHMP4B filament and main tube axis?
- Could embedding of the I-BAR protein at higher concentrations into the membrane influence CHMP2B binding?
- I am not convinced that the arrows in Fig. S3D point towards membrane structures. It rather looks like membrane-associated proteins, e.g. the CHMP filaments.
- The authors claim that CHMP4 spirals become disordered upon addition of CHMP2B. At least in the panel shown in Fig. 3C, the CHMP4 spiral seems to still have an ordered core as large as the entire spiral shown in Fig. 1A. Does the spiral core always stay ordered?
- Are the different classes of filaments obtained upon subtomogram classification mixed along tubes, or does one particular tube hold exclusively one kind of filament arrangement?
- Fig. S2C,D: Counts should be added.
- 'plunged frozen' should be 'plunge-frozen'
- It is almost impossible to recognize details in Fig. 3J – please include some close-up views.
- 'Ultrastructure' is normally rather used in the context of cellular structure. Maybe use 'arrangement' or 'architecture' instead?
- Which microscope setup do the imaging conditions in lines 533-535 correspond to? Please give conditions for the other setups, as well.
- "Tilted series" or "tilted-series" should be "tilt series"
- Line 544: the authors claim that they acquired tilt images with a constant dose of 0.8 e-/Å² for each tilt image. Is this true? This is rather unusual, because exposure is normally adjusted to the increasing sample thickness upon tilting, e.g. in a cosine like manner.
- Line 555: "defocus range between -1.5 and -4.25 mm" should certainly be "1.5 and -4.25 μm"

Reviewer #4 (Remarks to the Author):

In this study, Bertin and colleagues use a range of model membranes, designed to have positive or negative mean curvatures as well as regions of positive and negative Gaussian curvature and HS-AFM, fluorescence microscopy and cryo-EM/ET, to probe the curvature preferences of self-assembly of various ESCRT-III proteins and combinations.

The Snf7 homolog CHMP4B assembles into preferentially flat spirals and flattens curvatures of the membrane where they assemble, as verified by cryo-ET. This result is in contrast to recent studies that suggest Snf7 spirals buckle to induce curvature. Further data carefully demonstrates that CHMP4B does not form loaded-spring spirals and that, at least in the absence of other ESCRT-IIIs and with the lipid compositions used, CHMP4B has a preference for null curvatures.

Using laser-triggered liposome fusion, tube pulling and I-BAR treatment, the authors generate various membrane structures exhibiting regions of positive/negative and mean/Gaussian curvatures. These were coupled with encapsulation or external addition of various purified (and, in some cases, labeled) ESCRT-III proteins. CHMP2B alone colocalizes with regions of positive curvature. CHMP2A/3 together did not concentrate (or bind) to the interior of tubes. Again, they did bind positive curvature and, indeed, have a preference for this geometry.

Encapsulation of CHMP4B/CHMP2B or CHMP4B/CHMP2A/CHMP3 into vesicles where a tube had been pulled permitted assessment of whether these combinations could interact with regions of negative Gaussian curvature (the neck), a membrane geometry where ESCRT-IIIs have repeatedly been demonstrated to act *in vivo*. CHMP4B/2B bound to I-BAR-generated regions of positive curvature in these cases.

Finally, on noting that CHMP4B/2B together deform SUVs, cryo-EM/ET was used to study the deformations on LUVs. With the particular order of addition used, the ESCRT-III combinations deformed LUVs into pipe-surfaces. Sub-tomogram averaging was used on the deformed structures generated by CHMP4B/2B. Of the 3 classes where regular structures were displayed, one exhibited regular filaments arrayed in parallel around the circumference of the membrane, the smallest exhibited paired filaments and the last had filaments exhibiting perpendicular interconnections.

Overall, this is an insightful, elegant, meticulously executed and thorough dataset and I would strongly recommend publication. I have only minor comments for clarification or discussion.

- There appears to be a slight difference in burden of negatively-charged lipids used in CHMP4B SLB and LUV experiments (30% DOPS/10 % PIP2 vs 10% DOPS/10% PIP2). Is there a reason for this?
- Perhaps the authors could consider clarifying the methods, discussed in the text ~line 137: what is the composition of the uncharged GUV? I suppose the fusion lowers the PIP2 (and DOPS) to ~5% in each case. Is this correct?
- The fusion reaction decreases the salt concentration within the lumen of the fusing GUVs to ~50 mM. Altering salt concentration like this will of course change the affinity of charge interactions between protein and membrane, perhaps affecting extent of binding. As assembly of the ESCRT-III proteins is also essentially charge-based, it will also affect assembly. Could the authors comment on this to assuage this concern?
- Further to this, could the authors clarify if there is a difference in the salt environment with encapsulated ESCRT-III proteins and externally-added ESCRT-III used for binding assessments?
- Consider annotating the FT in fig S2C with scale, as it cannot be assessed at the moment whether the peaks are at the same inverse of resolution.
- Title for Tables 3 and 4 in the methods are duplicated
- Fig 3F appears to be a concentration-enhanced version of the projection image presented in 3E (instead of a zoom of the boxed area in 3I), as indicated in the text and legends

Rebuttal Letter

We thank very much the four reviewers for their questions and comments. We have addressed all points that were raised. In the following, our answers are in blue, the changes to the text in green and *italic*.

Reviewer #1

1) This paper by Bertin et al characterizes how ESCRT-III polymer assembly depends on local curvature of the membrane. The ESCRT field is a highly populated and - as far as I am aware - also quite contentious one, and some of the authors' conclusions appear in direct contradiction with previous claims by others. Following the editor's request, I herewith primarily comment on AFM aspects of the study, but not on the overall significance of the results, as that would be outside my area of expertise.

Overall, the manuscript is clearly written, but could be clarified in the Results section, lines 145-149, which describes to sorting ratios larger than 1 and smaller than 1, with references to Fig. 1F: Fig. 1F only shows sorting ratios smaller than 1. This text would benefit from clarification on what are expectations/predictions and what are results.

Lines 145-149 explain the general principle: the ratio is larger than 1 when proteins are enriched whereas it is lower than 1 in the case of depletion. It is correct that in Fig. 1F, the ratio is lower than 1; this is why we conclude that CHMP4B is excluded from the tube in these conditions.

We have corrected accordingly (page 5):

"Practically, this ratio is larger than 1 for proteins enriched and lower than 1 for proteins depleted from membrane tubes. This quantification reveals that over the large range of tube diameters that we could explore, since the sorting ratio is lower than 1, CHMP4B is mostly excluded from tubes with a negative mean curvature....."

2) Re AFM, the AFM results mostly appear complementary and not strictly essential to demonstrate the authors' points. The most intriguing observation is that of the formation of an outward protrusion at a vesicle upon addition of CHMP2B and CHMP4B, and the fact that they appear to show an intact vesicle on a mica substrate, whereas such vesicles tend to collapse on mica substrates. It would be helpful if the authors could comment (in main text of methods) what prevents the vesicles from collapsing on the mica. In addition, with such rather intriguing AFM results, it would be beneficial if an additional supplementary movie could be included, as a repeat to show the same (or a similar) result as acquired in an independent experiment.

This is an excellent point. Actually, we have used the same lipid composition for the GUV experiments and for the liposomes with AFM: 50.7 % EPC; 10% DOPS; 10% DOPE; 15 % cholesterol; 10% PI(4,5)P2; 0.2 % DSPE-PEG2000-Biotin; 0.1 % PE-Rhodamine. It turns out that, from our experience, cholesterol prevents fusion of the liposomes on the mica surface. This is why we have used a different composition, cholesterol-free, for the preparation of the SLBs (60% DOPC, 30% DOPS, and 10% PI(4,5)P2). In a separate manuscript where we have studied the effect of the ESCRT proteins on membrane mechanics, we have shown that we can apply indentation forces on the same type of liposomes, in the absence of proteins, without disrupting the membrane (see M. Alqabandi et al., bioRxiv, 756403 (2019)). The mechanical effect of cholesterol has been well documented in E. Evans work: he has shown that addition of cholesterol increases the stretching modulus of DOPC, or SOPC membranes (by at least a factor 3 for equimolar binary mixtures) as well as the lysis tension

(W. Rawicz ... E. Evans, Biophys. J. 94, 4725 (2008)), making them less prone to deformation and rupture.

A movie showing that our liposomes are very stable on mica can be downloaded from:

[http://xfer.curie.fr/get/ZVKDrywIX3e/Long lived vesicle on mica.avi](http://xfer.curie.fr/get/ZVKDrywIX3e/Long%20lived%20vesicle%20on%20mica.avi)

We have also added an additional movie showing budding in the presence of CHMP2B (New Supplementary Movie S5B). Eventually, we also have a new Supplementary Figure (Supplementary Fig. S7) showing other instances of budding as well as images of bare liposomes, or liposomes incubated with CHMP4B, at short and long time.

Reviewer #2

The paper by Bertin et al studies binding and assembly of ESCRTIII polymers on membranes of different curvatures. The study finds that CHMP4B prefers flat membranes, and in fact can flatten curved membranes, but cannot deform the membrane on its own. CHMP4B/CHMP2B and CHMP4B/CHM2A/CHMP3 both prefer positively curved membranes and can deform the membrane into positively curved helical tubes.

The structure and function of ESCRTIII polymers is currently a highly active field and this paper presents a very nice and timely addition to the field, which will be surely highly cited. The paper is clear and well written, and its results align with the recent findings of several other groups that are currently published as preprints, in particular Nguyen et al 10.1101/798181, Filseck et al 10.1101/716308, and Pfitzner et al 10.1101/718080. The previous papers deal with yeast ESCRTIII and CHMP1B+IST1, hence the current work is highly complementary. I would recommend publishing it in Nat Comm, but I would recommend addressing several points before:

We are very thankful to the reviewer #2 for her/his interest and very positive comments. We do agree that the work in these various preprints is very complementary to our own results and that it will be useful to the community to discuss them in this manuscript.

1) Can the authors say something about the stoichiometry of the proteins in these polymers? I think that would be extremely useful. Based on the previous work (eg Pfitzner et al), it is likely that the exact composition (including the stoichiometry) of the filament determines its curvature sensitivity and function. Hence could it be that the same combination of proteins, but at a different ratio, forms polymers that have a different curvature preference? Have the authors tested different relative concentrations? I think it might be misleading to claim that a combination of 2 proteins always does something, without specifying their ratio.

We thank the referee for this excellent point that makes sense when studying protein complexes.

First, in term of bulk concentrations, we have carefully screened the concentrations of the different proteins to promote optimal protein-membrane interaction and observe extensive tubulation with cryoEM. The protein concentration range leading to the “pipe-like” deformation without inducing any aggregation is rather narrow between 0.5 to 1 μ M (for CHMP4B). Above those concentrations most of the sample was aggregated while below no significant effect was observed. Similar concentrations have been used for our GUVs experiments, and 2 μ M for the AFM on LUVs.

Second, the stoichiometry of the proteins in the cryoEM experiments was: 1:1:3 for CHMP4B/CHM2A/CHMP3 and 1:1 for CHMP4B/CHMP2B. CHMP3 was used in large excess so that a significant amount of tubular structures could be observed. At a lower CHMP3 proportion, tubulation was reduced. CHMP3 seems to act as a nucleating factor which enhances tubulation, in the context of our experiments.

The use of excess CHMP3 is based on previous protocols to assemble CHMP2A-CHMP3 helical tubular structures *in vitro*. We do not consider an excess of CHMP3 problematic because CHMP3 is highly soluble and monodisperse across a wide range of concentrations. We never observed any CHMP3 polymers or aggregates of CHMP3 under physiological buffer conditions.

We therefore conclude that excess CHMP3 had no effect on the polymers formed here. Nevertheless, variations in the respective protein concentrations may affect the overall number of filaments, but not their structure and capacity to induce the helical membrane tubes. Notably, the same helical tubes have been induced with yeast proteins that form similar filaments (at least, at low resolution) but that have been employed at a different protein ratio, 2:1:1 (Snf7:Vps2:Vps24) and at much higher concentration (10 μ M of Snf7) as shown by von Filseck et al (J. M. von Filseck et al., bioRxiv, 716308 (2019)). This suggests that the pipe structure seems to form in many different conditions, including different stoichiometries.

Third, regarding curvature sensitivity and stoichiometry, our encapsulation method in the GUV experiments might lead to fluctuations of concentrations of the different proteins, in particular during the initial encapsulation step in the neutral GUVs. Nevertheless, we either observe no recruitment at all, or recruitment essentially at the tube neck, or on internal tubules with positive curvature. In general, we never have observed affinity for membranes with a negative mean curvature, either for the simple filaments or for the mixtures. It is possible that the degree of curvature sensitivity might change with stoichiometry, but it will not induce a drastic change of affinity for particular geometries.

2) In a similar paper Filseck et al comment how the membrane-binding face of filaments might be different between different polymers. Can the authors say something about this here?

As we mentioned in our manuscript, there is an essential difference between the paper from von Filseck et al and ours. Von Filseck and coll. use yeast proteins and observe very similar helical tubular structures, but they report that the ESCRT filaments form essentially pairs of double-stranded filaments with an asymmetric distribution around the tube, one class located on the equatorial part of the tube and the other on the polar part. They propose that to cope with the helical geometry of the tube, these two types of filaments should bind the membrane with different membrane-interacting surfaces. However, this type of organization with pairs of filament represents a very small fraction (3%) of the structures formed in our experiments with human ESCRT-III proteins. We rather observe single filaments periodically distributed around the tube (33% of the structures) (See Fig. S9A) without any clear difference on their binding face on the tube at the equatorial and polar position. We also have a large fraction (30%) composed of parallel filaments connected by perpendicular structures. Von Filseck and coll. use a ten-fold higher concentration of Snf7 (10 μ M) as compared to our experiments with CHMP4B (0.5 to 1 μ M). This might explain why we observe a very low fraction of double filaments, non-homogeneously distributed, and they don't. Nevertheless, we observe that the regularly-organized parallel filaments produce very similar helical tubes, in apparent contradiction with the theoretical model developed in von Filseck et al. that is based on the favored interaction of the polar filaments as compared to the equatorial ones. It shows that the versatility of the ESCRT-III proteins for binding membranes through different faces is not required to produce the pipe structure that is robustly formed by potentially different types of protein assemblies.

In the discussion part, we have added the following comments (page 10):

However, in this report, the global membrane shape transformation was essentially attributed to filament doublets non-homogeneously distributed around the tube, with different adhesion energy depending on the face in contact with the membrane.

..... Nevertheless, altogether our results and those from von Filseck et al⁶² show that the pipe shape constitutes a robust membrane deformation that occurs over a large range of protein concentrations, with proteins from different species.

3) Can the authors say something about the rigidity of the filaments, to attempt to explain the pipe structure?

We have shown that the fraction of tubulated structures does not increase when we incubate our lipid preparation with CHMP4B alone. Similarly, we did not observe any tubulation of GUVs, neither of LUVs when incubated with CHMP4B. It shows that CHMP4B polymers are probably too soft to deform membranes, but can nevertheless induce a twist in already-formed tubules. In other terms, it suggests that the bending rigidity of CHMP4B is lower than its torsional rigidity. Moreover, the preferred filament radius is probably large and positive according to our tube experiments. This is consistent with the discussion in von Filseck et al. ("We envision that the initial polymer made of single-stranded Snf7 is highly flexible and with a low spontaneous curvature to adapt to large membrane necks.").

We have added in the text (page 9):

Polymerization into spiral structures on the positively curved membranes of LUVs only leads to membrane flattening, but no membrane buckling occurs under these conditions in contrast to theoretical predictions^{22, 54}, suggesting that the filaments made of CHMP4B only have a low bending rigidity (see also⁶²).

This is different for the CHMP2A/CHMP3 polymers: we observe an increase by 2-fold of the tubular structures, which do not exhibit any twist. These results could be interpreted as these filaments having a higher bending rigidity than CHMP4B, with a stronger spontaneous curvature than CHMP4B since the filaments roll up around the tubes, but no significant torsional rigidity. Unfortunately, we cannot compare with other published data.

For CHMP2B, we have non-published data (M. Alqabandi et al., bioRxiv, 756403 (2019)) showing that the CHMP2B scaffold is elastically stiffer than that of CHMP2A/CHMP3. But, we never observed spontaneous tubulation of GUVs of LUVs with these proteins.

Eventually, the mixed complexes clearly have a significantly higher bending stiffness than the individual filaments, since we obtain 100% tubulation efficiency. As explained in point 2, Von Filseck et al. have developed a model for the buckling instability, based on different adhesion energies of filaments as a function of their binding face to the membrane. This favors the formation of thin helical tubes, in particular when tension is increased, rather of large straight tubes. We show here that the pipe structure can be obtained even when the filaments are regularly distributed, or when perpendicular structures are visible. In this last case, we can reasonably hypothesize that if filaments with a high spontaneous curvature C_0 and bending rigidity coexist with soft filaments with some torsional stiffness, it produces a helicoidal tubular structure where the tube radius is related to C_0 and a twist related to that of the soft filaments. In the other case, it is possible that the perpendicular structures are not regular enough to be detected in our reconstruction process, but can nevertheless crosslink the parallel filaments while creating local bending of the membrane. We also have observed that in one third of the cases, the global organization is too random to be analyzed; however, it leads to the same pipe structure. This suggests that the membrane deformation into a pipe shape is very robust and can result from potentially different protein organizations, but requires the presence of CHMP2A/CHMP3 or CHMP2B,

We have changed accordingly the corresponding paragraph in the discussion (page 10):

In addition, a large fraction of the surface is covered with single filaments regularly distributed around the tube diameter, without any apparent connection between them. It is possible that the connections between filaments are too scarce or too disorganized to be detected. Nevertheless,

altogether our results and those from von Filseck et al ⁶² show that the pipe shape constitutes a robust membrane deformation that occurs over a large range of protein concentrations, with proteins from different species.

4) It is interesting that Vps4 is not needed for the creation of deformation and the production of mechanical force. Can the authors comment on what its role could be?

First, Vps4 recycles ESCRT-III (Babst et al. EMBO J 1998). Secondly, Vps4 is required for finalizing membrane fission (Johnson et al eLife, 2019; plus many publications showing that dominant negative VPS4 blocks final steps, presumably in membrane fission). Thirdly a number of recent papers show that Vps4 actively remodels ESCRT-III *in vitro* (Maity et al, Sci Adv, 2019; Mierzwa et al Nat Cell Biol 2017) and *in vivo* (Mierzwa et al, Nat Cell Biol, 2017; Adell et al, eLife, 2017). Furthermore, VPS4-driven ESCRT-III remodeling is critical for successful HIV-1 release in order to complete budding (Bleck et al. PNAS 2014; Johnson et al. eLife 2018) indicating an active role in final steps of membrane remodeling including fission.

In addition, ATP-dependent remodeling/disassembly may prevent the formation of thermodynamically stable structures that could not produce any constructive constriction. Furthermore, a recent paper (L. Harker-Kirschneck et al BMC Biol. 17, 82 (2019)) proposes that the protein turn-over in the complex by Vps4 can induce a geometrical transition of the filaments (from a flat to a tilted spiral), leading eventually to fission in some circumstances. Here, our objective was to isolate the mechanical action of these ESCRT-III subunits and to study their affinity for specific geometries. Adding Vps4 would certainly be interesting but it is, however, beyond the scope of the present report. We prefer not to comment further on the role of Vps4 in our manuscript, since it would be too speculative.

5) In my opinion the authors did an excellent job in writing the paper in a clear manner, summarizing the findings at the end of the every section. However, I believe it is missing a connection with the currently published work. In particular with the Nguyen et al and Pfitzner et al. The community would greatly benefit from a comment on how it ties together.

Thanks for the suggestion. We now mention both preprints in our final discussion (page 10):

How these reciprocal interactions combined to the protein turn-over due to the VPS4 ATPase and the possible constrictive action of CHMP1 and INST1 ^{71, 72} lead to membrane scission remains to be established.

6) Along the same lines, it would be good to connect the more recent theoretical findings where the tense spiral hypothesis was questioned (Harker-Kirschneck et al, BMC Biology).

We thank the reviewer for mentioning this recent paper. We agree it should be discussed in our manuscript.

Accordingly, we have added in the discussion (page 10):

However, in this report, the global membrane shape transformation was essentially attributed to filament doublets non-homogeneously distributed around the tube, with different adhesion energy depending on the face in contact with the membrane. The authors propose that the possibility for the filaments to tilt and roll on the membrane for optimizing their binding can generate torque on the filament axis that can produce constriction and scission according to ⁷⁰.

7) In their conclusions the authors say this work provides new insight into how ESCRTIII assemblies generate force. But to me that is not obvious from this work, can the authors clarify this?

In the paragraph before this last sentence, we explained that the ESCRT proteins spontaneously assemble in filamentous structures that produce stresses on the membrane, which are reflected by the resulting pipe geometry. It is not a trivial shape and no published model has predicted this effect so far. *In vitro* experiments have reported only on the structure of the assembly of ESCRT-III proteins either in the absence of membrane or supported on solid substrates, which prevents to get information on induced membrane deformation. These stresses should also be present when the complexes are forced to assemble inside "necks" in biological situations; thus, the neck produced by upstream proteins (Gag, ESCRT-I/II etc..) should be remodeled by these forces. It is a mathematical/physical challenge to predict theoretically how membrane neck geometry would be affected by CHMP4B/CHMP2B or CHMP4B/CHMP2A/CHMP3 and this will certainly be the goal of future work to understand how it participates to neck constriction together with Vps4. Nevertheless, we believe that our work does reveal new aspects on the forces exerted by ESCRT-III assemblies.

Reviewer #3

I am serving as a technical reviewer for the cryo-electron tomography (cryo-ET) and subtomogram averaging experiments presented in this paper. As I am not an expert on ESCRT biology I cannot judge the novelty and the impact of the paper on the field.

The authors use a number of biophysical methods including cryo-ET to describe the interplay between polymerization of ESCRT-III components and membrane shape.

It is good practice in the cryo-EM field to deposit reported data for public access after publication. The authors should deposit at least one example tomogram and all of their subtomogram averages in the Electron Microscopy Data Bank (EMDB) and include the accession codes in the paper.

One example tomogram as well as our sub-tomogram averages have been deposited in the EMBD, with the accession code: EMD-10720. It will be made public in a year.

I appreciate that the authors visualized central aspects of their cryo-ET data in the form of very informative movies.

There are several aspects including the analysis and interpretation of the cryo-ET experiments that should be addressed in a revised version of the manuscript.

1) Major points:

a) Using AFM, the authors measure a peak-to-peak distance of filaments in CHMP4B spirals that is 50% smaller than what was observed using the same methods for the yeast homolog Snf7. Since this is not just 'slightly smaller' (line 110), the authors should discuss possible reasons.

We agree with the reviewer that the difference is significant. Our human construct is truncated on the C-terminus, which might lead to a different organization at the molecular level. In addition, we do not observe inter-filament branching as it was reported for Snf7 (N. Chiaruttini et al., Cell 163, 866 (2015)). These "bridges" between single filaments can increase the space between them.

We have replaced "slightly smaller" by "smaller" and added the following discussion (page 4):

Our human construct with a Δ C truncation might lead to a different molecular organization and the absence of branching might explain the difference we observe with the yeast protein.

b) The authors suggest that CHMP4B filament spirals flatten vesicles. This is a very problematic conclusion, because isolated vesicular structures always appear significantly flattened in cryo-ET data, most likely because they are squeezed by the surface tension of the water/buffer film during the blotting process that is part of the cryo-EM grid preparation. There are numerous published cryo-

ET studies that document this effect. In the methods section, the authors even state that vesicles are typically embedded in “a thin (less than 100 nm) film of aqueous solution”, which would clearly compress them in Z-direction. While I agree that CHMP4B seems to colocalize with flat membrane areas, there is no experimental support that CHMP4B actually induces the flattening. It appears more likely that the flattening is a result of the surface tension and CHMP4B preferentially co-localizes with these areas. The authors should discuss this possibility in the manuscript and possibly tone down their claim that CHMP4B filaments are the cause for flattened vesicles.

As the referee mentioned, the vitrification process within a thin film of solution might partly induce some flattening of the vesicles. To minimize this effect, we have used lacey grids rather than quantifoil grids, on purpose. Indeed, lacey grids are heterogenous and the size of the holes (from 50 nm to several microns in diameter) can accommodate from small to large vesicles without inducing drastic deformations as visualized on a control image (Figs. S3A and B). Besides, we have generated liposomes from re-suspended dried lipid films. This methodology produces deformable vesicles heterogeneous in size, from 50 nm to 1 um in diameter. Hence the thin film we refer to in the manuscript is thicker than 100 nm to accommodate bigger vesicles and visualize any deformations. As shown below in the answer to question c), our cryo-tomograms show that the interaction of the liposomes induces a flattening of the bare vesicles.

Nevertheless, we would like to stress that our main message is that CHMP4B does not induce membrane tubulation as previously predicted.

c) How can the authors judge vesicle flattening in 2D projection images as shown in Fig. S3A,B and where are the control vesicles in Fig. S3A,B that the authors must have compared the CHMP4-decorated vesicles to in order to come to their conclusion?

This is a fair comment. To address it, in addition to providing 2D images where round-shaped vesicles are visible (Figs. S3A and B), we have carried out cryo-tomography on control vesicles (see new Supplementary Movies S2A-B). When measuring the height of control vesicles, we find a thickness of about 150 nm. Note that the top and bottom of the bare vesicles, (i.e. lipid bilayers which are in average perpendicular to the electron beam) cannot be distinguished because of the missing wedge. However, when the CHMP4B spirals are bound to the surface of liposomes (top and bottom), the height of the vesicles decreases down to about 50 nm, suggesting that the proteins significantly squash the liposomes.

We have added the following sentence in the revised version (page 4):

When bound to membranes, CHMP4B spirals (in red) are clearly flat and follow the contour of vesicles without inducing any noticeable deformation as highlighted in the side view (Fig. 1C, bottom) and in Supplementary Movie S3. They induce a squashing of the vesicles with a height that decreases from about 150 nm for the bare liposomes (Supplementary Movies S2A-B) down to about 50 nm with bound CHMP4B spirals (Supplementary Movie S3). This suggests that the elastic energy stored in the CHMP4B spirals favors a non-curved membrane and no invagination or ex-vagination.

d) In the CHMP2A/CHMP3 binding experiment, the authors specifically state that ‘proteins were reconstituted inside GUVs at nearly physiological concentrations’. What about concentrations in the other experiments?

The protein concentrations used for the reconstitution inside GUVs are reported in Table 5. We removed the statement “at nearly physiological conditions”, which is misleading since we do not know the exact physiological concentrations of ESCRT-III proteins in cells.

We therefore replaced the sentence by : *“proteins were reconstituted inside GUVs at low micromolar concentrations”*.

The concentrations used for the AFM or cryo-EM experiments are similar, around 0.5-1 μM . We also realized that the concentrations for the Cryo-EM experiments were missing in our previous manuscript. This has been fixed.

e) And related: CHMP concentration seems relevant to membrane binding (line 259). How much does the concentration bias the experiments?

A current hypothesis is that ESCRT-III activation leads to polymerization concomitant with membrane binding *in vivo*. Thus, the concentration is important for function, if the nucleation of polymerization depends on the local concentration *in vivo* and not on other cellular factors. We favor that the concentration is important and determines the extend of membrane binding and remodeling as observed *in vitro*.

See also the response to question 1 from reviewer 2 (a similar question): The protein concentration range leading to the described “pipe-like” deformations without inducing any aggregation (which occurs at higher concentrations) is rather narrow. The concentrations range from 0.5 to 1 μM for CHMP4B, CHMP2B and CHMP2A. Below, pipe structures are rarely detected, suggesting that there is an optimum to observe it.

We have mentioned this result in the manuscript (page 7):

“ Close to 100% of the LUVs were tubulated under these conditions. This concentration range is optimal since no extensive tubulation is observed below, and protein aggregates form above.”

f) The authors claim that the diameter of CHMP4B-decorated vs. CHMP4B-CHMP2B-decorated tubes is significantly larger (line 317). At least judging from the size of the error bars in Fig. 3L, this seems not to be the case.

We have calculated the p value to show that the difference is indeed significant and found $4,5 \cdot 10^{-6}$ and $7,4 \cdot 10^{-5}$ for CHMP4-CHMP2B and CHMP4-CHMP2A-CHMP3, respectively. They have been added on Fig. 3L.

g) I think the entire subtomogram averaging section of the paper could benefit immensely from some further analysis. For instance, single filament subunits are not resolved along the tube axis and the bridges in Fig. 4C are not convincing. Given that the raw data acquired at EMBL in Heidelberg was likely of very high quality, one possible cause for the rather low resolution obtained during subtomogram averaging could be limited long-range order, in particular regarding the distance between individual filaments. Therefore, the authors could try to focus processing on smaller segments using tighter masks e.g. excluding not directly adjacent filaments.

We fully agree with the reviewer's suggestions and work to improve the reconstructions is currently underway. However, progress is slow and because it may take many more months, we believe that it is beyond the scope of the current manuscript. In fact, the goal of this manuscript is to show the relationship between ESCRT membrane curvature sensitivity, the global architecture of ESCRT filaments bound to membranes and the resulting membrane deformations.

h) Related to the previous comment: The FSC in Fig. S8B never goes down to zero! This should never happen and needs to be resolved. The most likely cause for this FSC behavior is the presence of duplicate particles in the dataset, which result in artificially high correlation and thus overestimated resolution.

We thank the referee for noticing this issue. We have revisited our data processing and generated FSCs calculated using soft masks instead of hard masks that now decay to zero. Figure S8 (now Fig. S9) has been modified accordingly. In the Fig. R1 below, we show how the FSCs are modified using different masks.

Figure R1: FSCs generated using no mask (grey), hard masks (dotted dashed line) or soft masks (hard line) for single filaments (top left), doublets of filaments (top right) or networks of filaments (bottom).

The resolutions thus calculated using the FSCs at a 0.5 threshold are 26.1 Å for both single and doublets of filaments and 28.3 Å for networks of filaments, respectively. We have corrected the values in the revision.

2) Minor points:

a) In many of the electron micrographs or tomograms, the contrast is rather low and the features discussed in the text are hardly visible. I guess the authors are showing 'raw' unfiltered data. The contrast of features could be improved by averaging several Z-slices or by filtering the data, e.g. with a low-pass filter, a Wiener filter or denoising algorithms.

The displayed slices from tomograms were already averaged from several slices and filtered as suggested to enhance the contrast, but we cannot do better. On individual images, we have plotted side-by-side images with guidelines for the eyes to help visualization.

b) Does the behavior of CHMP4B in the binding experiments in Figs. 1E,F differ from what has been observed for the yeast homolog Snf7? If so, could the authors repeat some of the binding experiments with Snf7 to confirm the different behavior?

A preprint from A. Roux's group (A.-K. Pfitzner, V. Mercier, A. Roux, bioRxiv, 718080 (2019)) shows that Snf7 is unable to bind inside tubes pulled from GUVs, below a threshold radius of 115 nm. This is fully consistent with our results in Figs. 1E, F. Thus, we did not replicate these experiments.

c) The authors claim that CHMP4B filaments “tend to be aligned along the main tube axis”. Could the authors confirm this in a more objective manner for instance by plotting a histogram of the angular difference between CHMP4B filament and main tube axis?

We thank the reviewer for stressing this point. On deformable tubes, we do see that the filaments seem to remain strictly parallel to the tube axis (see Fig. 1I). In contrast, on rigid lipid tubes, the filaments deviate slightly from the tube axis since they display an intrinsic curvature. We have carried out the suggested analysis and confirmed that indeed CHMP4 filaments tend to align, on average, parallel to the axis of the rigid tubes. We find an average angle of 8.2 degrees +/- 5.1 degrees (SD, N=21). We have added this measurement to the main text.

d) Could embedding of the I-BAR protein at higher concentrations into the membrane influence CHMP2B binding?

In our experiments, the I-BAR protein is incubated *outside* of the GUV (which induces invaginations), whereas CHMP2B is encapsulated *inside* it. I-BAR concentrations only influence the degree of tubulation of the GUV (Z. Chen, Z. Shi, T. Baumgart, Biophys. J. 109, 298 (2015)), but since both proteins are not on the same side of the membrane, it should not have any effect on CHMP2B binding.

e) I am not convinced that the arrows in Fig. S3D point towards membrane structures. It rather looks like membrane-associated proteins, e.g. the CHMP filaments.

Indeed, the reviewer is right and we thank him for noticing this mistake in the main text. We originally intended to state that the arrows point towards filamentous ESCRT proteins. This has been corrected in the main text (*They produce short tubular structures (tens of nanometers long) on spherical liposomes (Supplementary Fig. S3D, N = 2 experiments) and in the figure legend (CHMP2A-ΔC/CHMP3 form short tubular structures (white arrows) extending out of the vesicle.)*)

f) The authors claim that CHMP4 spirals become disordered upon addition of CHMP2B. At least in the panel shown in Fig. 3C, the CHMP4 spiral seems to still have an ordered core as large as the entire spiral shown in Fig. 1A. Does the spiral core always stay ordered?

The most striking effect upon addition of CHMP2B is that the spirals become much less regular, with large gaps between turns, as visible in Supplementary Figs. S6A (line 2) or in the new panel G of Supplementary Fig. S6 that shows a few examples of distorted spirals.

For certain spirals, we see a large core that remains more or less unaltered and for others a smaller core. We have added a new panel (F) to Supplementary Fig. S6 that shows snapshots of Supplementary Movie S4, where it can be seen that the core initially relatively small becomes larger over time. Supplementary Figs. S2D and S6E also show that, when CHMP2B is added, the diameter of the core becomes larger in average (28.6 to 40.6 nm) with a larger spreading.

g) Are the different classes of filaments obtained upon subtomogram classification mixed along tubes, or does one particular tube hold exclusively one kind of filament arrangement?

We were not clear enough on this point. Each tube exhibits exclusively one kind of filament arrangement. We do not see mixed arrangements within the same “pipe”.

We have now clarified this point in the text (page 8):

Four different populations of filaments decorating the pipe-like architecture could be identified in the whole dataset. Note that each tube exhibits exclusively one type of filament arrangement and no mixed arrangements coexist within the same “pipe”.

We have realized that the structure schematized on the Fig. 4C bottom can be misleading, since it shows coexistence of zones with "single" filaments with zones containing orthogonal structures. We have changed this figure that now exhibits one type of structure only on a tube.

We also removed the end of the paragraph in the discussion about a co-existence of structures that was not correct:

~~Continuity of the surface and of the filaments may explain why the tubular geometry is maintained in the absence of connections between the spiraled filaments.~~

h) Fig. S2C,D: Counts should be added.

The numbers of samples were provided in the figure legend in the initial version. We keep them in the legend.

i) ‘plunged frozen’ should be ‘plunge-frozen’

Done

j) It is almost impossible to recognize details in Fig. 3J – please include some close-up views.

We did a mistake when assembling the different images for Figure 3: Fig. 3J was a duplicate of Fig. 3E. We apologize for this. We have replaced it by the intended image (an enlargement from Fig. 3I), which displays details unambiguously.

k) ‘Ultrastructure’ is normally rather used in the context of cellular structure. Maybe use ‘arrangement’ or ‘architecture’ instead?

The text has been changed accordingly.

l) Which microscope setup do the imaging conditions in lines 533-535 correspond to? Please give conditions for the other setups, as well.

The imaging conditions for each microscope and setup are now specified in the Methods section. The set-up corresponding to lines 533-535 is the Grenoble set-up. We have clarified this point in the revised paper.

m) “Tilted series” or “tilted-series” should be “tilt series”

This has been corrected

n) Line 544: the authors claim that they acquired tilt images with a constant dose of 0.8 e-/Å² for each tilt image. Is this true? This is rather unusual, because exposure is normally adjusted to the increasing sample thickness upon tilting, e.g. in a cosine like manner.

The dose was not adjusted with tilting. It remained constant throughout data collection. However, we chose to collect data using a symmetric dose scheme described in Hagen et al. (2017, J. Struct. Biol.) Indeed, the high-resolution information is enclosed within the lower tilts where the sample is thinner and this procedure is known to be crucial to pursue successful sub-tomogram averaging.

o) Line 555: “defocus range between -1.5 and -4.25 mm” should certainly be “1.5 and -4.25 μm ”
We thank the reviewer for pointing out this typographical error and we have replaced mm by μm .

Reviewer #4

In this study, Bertin and colleagues use a range of model membranes, designed to have positive or negative mean curvatures as well as regions of positive and negative Gaussian curvature and HS-AFM, fluorescence microscopy and cryo-EM/ET, to probe the curvature preferences of self-assembly of various ESCRT-III proteins and combinations.

The Snf7 homolog CHMP4B assembles into preferentially flat spirals and flattens curvatures of the membrane where they assemble, as verified by cryo-ET. This result is in contrast to recent studies that suggest Snf7 spirals buckle to induce curvature. Further data carefully demonstrates that CHMP4B does not form loaded-spring spirals and that, at least in the absence of other ESCRT-IIIs and with the lipid compositions used, CHMP4B has a preference for null curvatures.

Using laser-triggered liposome fusion, tube pulling and I-BAR treatment, the authors generate various membrane structures exhibiting regions of positive/negative and mean/Gaussian curvatures. These were coupled with encapsulation or external addition of various purified (and, in some cases, labeled) ESCRT-III proteins. CHMP2B alone colocalizes with regions of positive curvature. CHMP2A/3 together did not concentrate (or bind) to the interior of tubes. Again, they did bind positive curvature and, indeed, have a preference for this geometry.

Encapsulation of CHMP4B/CHMP2B or CHMP4B/CHMP2A/CHMP3 into vesicles where a tube had been pulled permitted assessment of whether these combinations could interact with regions of negative Gaussian curvature (the neck), a membrane geometry where ESCRT-IIIs have repeatedly been demonstrated to act in vivo. CHMP4B/2B bound to I-BAR-generated regions of positive curvature in these cases.

Finally, on noting that CHMP4B/2B together deform SUVs, cryo-EM/ET was used to study the deformations on LUVs. With the particular order of addition used, the ESCRT-III combinations deformed LUVs into pipe-surfaces. Sub-tomogram averaging was used on the deformed structures generated by CHMP4B/2B. Of the 3 classes where regular structures were displayed, one exhibited regular filaments arrayed in parallel around the circumference of the membrane, the smallest exhibited paired filaments and the last had filaments exhibiting perpendicular interconnections.

Overall, this is an insightful, elegant, meticulously executed and thorough dataset and I would strongly recommend publication. I have only minor comments for clarification or discussion.

1) There appears to be a slight difference in burden of negatively-charged lipids used in CHMP4B SLB and LUV experiments (30% DOPS/10 % PIP2 vs 10% DOPS/10% PIP2). Is there a reason for this?

As we explained to reviewer 1 (point 2) we had to adjust our lipid composition to be able to fuse liposomes and prepare SLBs: we had to remove cholesterol and add more charged lipids.

2) Perhaps the authors could consider clarifying the methods, discussed in the text ~line 137: what is the composition of the uncharged GUV? I suppose the fusion lowers the PIP2 (and DOPS) to ~5% in each case. Is this correct?

This is perfectly correct. One GUV is not charged and the other contains 10% PIP2 and 10 % DOPS. After fusion, the final charge is lower and depends of the relative size of each GUVs, but in general it should be divided by about a factor 2, since the considered GUVs have more or less the same size. The precise compositions are provided in the Methods section.

We have added the following sentence to clarify this point (page 4):

We have encapsulated fluorescently-labelled CHMP4B inside a non-charged GUV (cyan) at a concentration of about 1 μ M and fused it with a GUV containing PI(4,5)P2 (magenta) (Fig. 1E) (for the details on lipid compositions, see Methods), leading to a reduction of the charge on the final GUV of about a factor 2.

3) The fusion reaction decreases the salt concentration within the lumen of the fusing GUVs to ~50 mM. Altering salt concentration like this will of course change the affinity of charge interactions between protein and membrane, perhaps affecting extent of binding. As assembly of the ESCRT-III proteins is also essentially charge-based, it will also affect assembly. Could the authors comment on this to assuage this concern?

This is correct. We have provided the final salt and protein concentrations in the Tables 1 to 5. In all GUV experiments, either after fusion with protein encapsulated or with proteins outside the GUVs, the salt concentration is about 50 mM. It is also the case for the experiments on SLBs with hs-AFM, or for the cryo-EM with the LUVs. Because refolded CHMP2B full length and delta C are sensitive to NaCl concentrations above 100 mM, we adjusted the experiments to these conditions.

4) Further to this, could the authors clarify if there is a difference in the salt environment with encapsulated ESCRT-III proteins and externally-added ESCRT-III used for binding assessments?

As already detailed in the Methods section, the final salt concentration inside the GUVs or when ESCRTs are externally-added is about 50 mM.

To make it clear, we have specified this point in the text (page 14):

- *maintaining a final NaCl concentration of ~50mM after fusion (each protein is stored in a different storage buffer and at a different concentration), similar to the experiments with binding outside of the GUVs;*

5) Consider annotating the FT in fig S2C with scale, as it cannot be assessed at the moment whether the peaks are at the same inverse of resolution.

The reviewer probably meant Fig. S3C. A scale bar of 1 nm⁻¹ has been added, as requested.

-6) Title for Tables 3 and 4 in the methods are duplicated

Thanks for noticing this. It has been changed accordingly.

7) Fig 3F appears to be a concentration-enhanced version of the projection image presented in 3E (instead of a zoom of the boxed area in 3I), as indicated in the text and legends.

We understand that the reviewer has been confused by Fig. 3. In assembling the different images for Figure 3, we duplicated Fig. 3E in Fig. 3J. We apologize for this mistake. We have replaced Fig. 3J by the intended image (an enlargement of Fig. 3I), which unambiguously displays details.

REVIEWERS' COMMENTS:

Reviewer #1 (Remarks to the Author):

The authors have fully addressed my comments on this manuscript (which pertained to the AFM-technical aspects of it).

Reviewer #2 (Remarks to the Author):

I have carefully read the revised version of the manuscript by Bertin et al. and I can confirm that the authors have addressed most of my comments. I appreciate their clarification on the structures, and comparing their work with recent results from other groups. I agree with their comment in the reply that the orthogonal structures are likely needed to induce the pipe shape, as it's not obvious how straight filaments alone would cause this. Perhaps this would be worth briefly commenting on in the text. It is also possible that the orthogonal structures have different protein composition than the straight ones.

In any case, to reiterate, I enjoyed reading this paper and have found it a valuable addition to the field, and am looking forward to seeing it published.

Reviewer #3 (Remarks to the Author):

In their revised manuscript Bertin et al. have addressed all of my concerns and suggestions. They have included several additional analyses that strengthen the paper and make it suitable for publication in Nature Communications.

Reviewer #4 (Remarks to the Author):

The revised manuscript by Bertin, de Franceschi and others has made some selective and judicious edits compared to the original that enhance the work. Particularly welcome are the additional discussion of the other recent contributions by the Frost and Roux labs that put this work in context for the reader. I remain enthusiastic about the insights generated by this combination of in vitro work with templates of varying geometries and cryo-EM and cryo-ET and continue to recommend publication.

One issue still requires clarification. In the manuscript, the authors state (line 95) that full-length CHMP4B is used in this work. The methods describe purification of CHMP4B (line 527) where the purification protocols of CHMP2A and B note they are C-terminal truncations. The authors respond to the comment of reviewer 3 about the CHMP4B spiral peak to peak distance as being smaller than that of Snf7 by noting that a del C construct is here used. Is this a CHMP4B del C construct? Does this work therefore have a mixture of truncated and non-truncated CHMP4B at various points? If it is full length, the explanation given for the smaller peak to peak distance needs to be improved. If it is del C, additions and clarifications need to be made throughout, and appropriate modifications made to the methods.

Rebuttal Letter

We sincerely thank the reviewers for their very thorough and constructive reports and for taking the time to review our manuscript in such difficult times.

Reviewers #1 and #3 had no further requests:

Reviewer #1

The authors have fully addressed my comments on this manuscript (which pertained to the AFM-technical aspects of it).

Reviewer #3

In their revised manuscript Bertin et al. have addressed all of my concerns and suggestions. They have included several additional analyses that strengthen the paper and make it suitable for publication in Nature Communications.

Our replies to Reviewers #2 and #4 are in blue

Reviewer #2

I have carefully read the revised version of the manuscript by Bertin et al. and I can confirm that the authors have addressed most of my comments. I appreciate their clarification on the structures, and comparing their work with recent results from other groups. I agree with their comment in the reply that *the orthogonal structures are likely needed to induce the pipe shape, as it's not obvious how straight filaments alone would cause this. Perhaps this would be worth briefly commenting on in the text. It is also possible that the orthogonal structures have different protein composition than the straight ones.* In any case, to reiterate, I enjoyed reading this paper and have found it a valuable addition to the field, and am looking forward to seeing it published.

Thanks for the suggestion. We have stressed further the differences between our manuscript and that of von Filseck et al. (Ref. 63) and also the possibility that the compositions of the orthogonal filaments might be different.

New text (addition in green) in the discussion section

...The authors propose that the possibility for the filaments to tilt and roll on the membrane for optimizing their binding can generate torque on the filament axis that can produce constriction and scission according to 70. This type of structure however represents only a very minor fraction of the organizations that we have observed with human ESCRTs, **suggesting that other mechanisms can also shape vesicles in to pipe surfaces.....** This sort of scaffold that combines both trends of CHMP4B to form a wide spiral organization and of CHMP2A/CHMP3 to wrap around tubes, can explain the emergence of the pipe surface geometry for the membrane, although what sets the tube diameter is unclear. **The respective protein compositions of the perpendicular structures might also be different, but at this stage we cannot distinguish them...**

Reviewer #4

The revised manuscript by Bertin, de Franceschi and others has made some selective and judicious edits compared to the original that enhance the work. Particularly welcome are the additional discussion of the other recent contributions by the Frost and Roux labs that put this work in context for the reader. I remain enthusiastic about the insights generated by this combination of in vitro work with templates of varying geometries and cryo-EM and cryo-ET and continue to recommend publication.

One issue still requires clarification. In the manuscript, the authors state (line 95) that full-length CHMP4B is used in this work. The methods describe purification of CHMP4B (line 527) where the purification protocols of CHMP2A and B note they are C-terminal truncations. The authors

respond to the comment of reviewer 3 about the CHMP4B spiral peak to peak distance as being smaller than that of Snf7 by noting that a del C construct is here used. Is this a CHMP4B del C construct? Does this work therefore have a mixture of truncated and non-truncated CHMP4B at various points? If it is full length, the explanation given for the smaller peak to peak distance needs to be improved. If it is del C, additions and clarifications need to be made throughout, and appropriate modifications made to the methods.

We thank the reviewer very much for noticing this inconsistency. We only use the truncated version of CHMP4 in the whole work. We have corrected this mistake both in the Introduction, and in the Methods section. We also have corrected the figures and the legends accordingly

New text:

...We use C-terminally truncated versions of CHMP4B, CHMP2A and CHMP2B to facilitate polymerization as well as full-length CHMP4B and CHMP3...

CHMP4B- Δ C, containing residues 1-175 followed by a linker of 20 aa

(SNSASDDASASASADEDASS) and CHMP4B residues 204-224, was expressed as MBP.....